# A mutation in switch I alters the load-dependent kinetics of myosin Va

Christopher Marang[1], Brent Scott[1], James Chambers[2], Laura K. Gunther[3], Christopher M. Yengo[3] & Edward P. Debold[1] ✉

Myosin Va is the molecular motor that drives intracellular vesicular transport, powered by the transduction of chemical energy from ATP into mechanical work. The coupling of the powerstroke and phosphate ($P_i$) release is key to understanding the transduction process, and crucial details of this process remain unclear. Therefore, we determined the effect of elevated $P_i$ on the force-generating capacity of a mini-ensemble of myosin Va S1 (WT) in a laser trap assay. By increasing the stiffness of the laser trap we determined the effect of increasing resistive loads on the rate of $P_i$-induced detachment from actin, and quantified this effect using the Bell approximation. We observed that WT myosin generated higher forces and larger displacements at the higher laser trap stiffnesses in the presence of 30 mM $P_i$, but binding event lifetimes decreased dramatically, which is most consistent with the powerstroke preceding the release of $P_i$ from the active site. Repeating these experiments using a construct with a mutation in switch I of the active site (S217A) caused a seven-fold increase in the load-dependence of the $P_i$-induced detachment rate, suggesting that the S217A region of switch I may help mediate the load-dependence of $P_i$-rebinding.

Myosins are a family of molecular motors responsible for a multitude of motions and forces within a cell, including tasks such as muscle contraction[1] and vesicular transport[2]. Myosin Va is a prototypical processive myosin which steps along the 36 nm pseudo-repeat of an actin filament[3,4] to transport its cargo to specific locations within the cell. Myosins most often work together in teams to accomplish their intracellular tasks[5–7]. Even processive motors such as myosin Va are now thought to drive cargo transport as multiple motor teams[6], thus it is important to understand how ensembles of myosin generate force and motion. Common to all myosins is the ability to transduce the chemical energy from ATP into force and motion[8], enabling them to accomplish a myriad of cellular tasks. Indeed the motor domain of myosins, which includes the nucleotide-binding site and the actin-binding domain is highly conserved across the myosin family[9]. While these aspects are well known the mechanism underlying the motor domain's ability to couple the release of chemical energy from ATP with the generation of force and/or motion is still unclear[10–14]. This gap in knowledge limits the understanding of how this superfamily of molecular motors accomplishes a multitude of intracellular tasks, which in turn limits our ability to identify the appropriate processes and structures to target for the development of effective treatments for a myriad of myosin-associated diseases[15,16].

The basic sequence of steps in myosin's chemo-mechanical cycle are well-established[17,18], for example, binding of ATP to myosin's nucleotide-binding site releases it from its molecular track, actin. While detached from actin, myosin hydrolyses the ATP to ADP and $P_i$. Hydrolysis of the ATP is thought to power the re-priming[19] of the long alpha helix, which serves as a lever arm, enabling production of the next force-generating powerstroke. The most crucial, but most poorly characterized, step in the transduction process is the coupled release of $P_i$ and the powerstroke[10,12,20,21], this is therefore key to elucidating the mechanism of energy transduction by myosin[22]. In order to understand these steps investigators typically bathe myosin or muscle in excess amounts of $P_i$ to understand how it might rebind to the nucleotide-

[1]Department of Kinesiology, University of Massachusetts, Amherst, MA 01003, USA. [2]Institute for Applied Life Sciences, University of Massachusetts, Amherst, MA 01003, USA. [3]Department of Cellular and Molecular Physiology, Penn State College of Medicine, Hershey, PA 17033, USA. ✉e-mail: edebold@umass.edu

binding site and putatively reverse the $P_i$-release step, and the powerstroke. Early observations using isolated myosin in solution showed that actomyosin with ADP bound in it active site (AM.ADP) has a very low affinity for $P_i$[17]. Consistent with this observation, elevated levels of $P_i$ have little effect on unloaded shortening velocity in skinned muscle fibers[23,24]. Crystallization of myosin bathed in high amounts of $P_i$[21] demonstrates that $P_i$ does not appear in the active site unless it is exposed to very high $P_i$ concentrations (70–100 mM) for long periods of time (45 min). Thus, these observations suggested that $P_i$ does not readily rebind to myosin's active site, and therefore suggested that the release of $P_i$, and the powerstroke are largely irreversible[25]. However, elevated levels of $P_i$ strongly depress isometric force in muscle fibers, even at rather low $P_i$ concentrations[24,26,27], suggesting that $P_i$ can readily rebind to myosin's active site in these preparations. A key difference between the experiments used to reach these disparate conclusions was the amount of resistive load experienced by the myosin. In solution experiments, or in crystallography preparations, the amount of load opposing myosin's powerstroke is thought to be negligible. By contrast, during an isometric contraction of muscle, myosin is thought to experience a high degree of resistive force, i.e. force opposing the direction of the powerstroke[7,28]. Based on these and similar findings it is believed that an external resistive load can modulate the affinity of actomyosin for $P_i$ while in an ADP-bound state[26,27,29]. However, the load-dependence of $P_i$ rebinding to myosin, and more specifically, to actomyosin in an AM.ADP state, has not been fully quantified at the molecular level. This is crucial information for understanding exactly how the force-generating powerstroke and the release of $P_i$ from the active site are coupled[22] and thus is critical to understanding how this prototypical molecular motor transduces chemical energy into mechanical work[12].

It is also unclear which regions or elements within myosin's active site might mediate the load-sensitivity of $P_i$-rebinding to myosin's active site[10,12,20]. The release of $P_i$ is thought to occur through an opening created by a transient conformational change in the key, switch II, element of the nucleotide binding site, referred to as the back-door exit[30]. Presumably, when $P_i$ rebinds to actomyosin (in an AM.ADP state) it reverses this pathway and thus by studying re-binding the pathway of $P_i$-release can be revealed[21]. Serine 217, within the switch I element of the active site, is thought to play a key role in the release of $P_i$ from myosin's nucleotide binding site[31]. This residue putatively makes contacts with, and guides, the release of the gamma-$P_i$ of ATP out of the active site[32]. Support for this role is provided by the observations that an S217A construct displays a decreased rate of $P_i$-release[21,31,33]. Thus, manipulating this residue presents an ideal opportunity to potentially alter the dynamics of $P_i$-release and rebinding in the presence of high levels of $P_i$ and a high resistive load, providing a window into the nature of the crucial coupling between force-generation and $P_i$-release.

In this work we characterized the load dependence of $P_i$-rebinding to a mini-ensemble of myosin Va motors in a three-bead laser trap assay, using WT and S217A constructs of single-headed myosin Va. This was done to provide insight into the mechanisms involved in the coupling between the powerstroke and $P_i$-release, and rebinding to myosin's active site. The findings demonstrate that the $P_i$-induced detachment rate increases with increasing load, and that this load sensitivity is greatly increased in the S217A construct. We also show that despite $P_i$ rebinding to the active site and inducing detachment from actin, WT myosin maintains the ability to displace the actin filament, consistent with the powerstroke occurring prior to the release of $P_i$ from the active site.

## Results

To gain insights into how myosin couples force-generation and the release of $P_i$ from the active site we characterized the load-dependence of $P_i$-rebinding to a small team, or mini-ensemble, of single-headed (S1) myosin Va molecules using a three-bead optical trapping assay (Fig. 1a). The number of myosins available to interact with the actin filament was determined by directly counting of the number of myosin molecules on the coverslip surface using Stochastic Optical Reconstruction Microscopy (STORM)[34] in a parallel set of experiments (see Methods). These data were combined with the defined geometry of the three-bead laser trap assay to determine the length of the actin filament available to the myosin molecules (Supplementary Fig. 1D). The results indicated that, at the myosin concentration used, an average 4-5 molecules were available to interact with the single actin filament in the three-bead laser trap assay (Supplementary Fig. 1).

### Increased trap stiffness accelerates $P_i$-induced detachment

In the absence of added $P_i$ multiple myosin molecules bound to, and translocated, the actin filament against the spring-like load of the laser trap (Fig. 1b). Thus, the load on the myosins progressively increased as the bead-actin-bead assembly was displaced from the center of the laser trap (Fig. 1b). We quantified the duration of these interactions by counting the time from the first attachment to the last detachment (Fig. 1b), using a mean-variance-threshold algorithm to detect individual binding events (see Methods).

Increasing the stiffness of the laser trap from 0.04 to 0.1 pN. $Nm^{-1}$ caused the binding event lifetimes to significantly increase by roughly 50% (Fig. 2a). This is consistent with the increased resistive load slowing the rate of ADP-release from myosin's active site[35–37]. Repeating these experiments in the presence of 30 mM $P_i$ caused the opposite effect resulting in visibly shorter binding event lifetimes, that is apparent in the raw displacement records (Fig. 1b). Indeed, in the presence of 30 mM $P_i$ a full analysis of the full data set revealed that the average event lifetime was reduced by 25% (Fig. 2a). Increasing the stiffness of the laser traps from 0.04 to 0.06 pN.$nm^{-1}$ caused a further reduction in the event lifetimes reducing them by another 15% (Fig. 2a). This pattern continued at the highest trap stiffness (0.10 pN.$nm^{-1}$), where the binding event lifetimes were reduced to less than 50% of the control value (0.04 pN.$nm^{-1}$ and 0 mM $P_i$). The effect on the complete distribution of binding events is evident in the cumulative distributions (Fig. 2b). Specifically, the introduction of 30 mM $P_i$, and then the increased laser trap stiffness caused a strong leftward shift in the plots, indicative of an increase in the number of shorter duration binding events. A similar pattern emerges from the histograms of the binding event lifetimes, where the longest events are eliminated with added $P_i$ and increasing trap stiffness, shifting the center of the distribution leftward (Insets, Fig. 2b).

### S217A increases load sensitivity of $P_i$-induced detachment

In contrast to the findings using WT myosin Va, when these experiments were repeated with the S217A construct it showed that elevated $P_i$ had almost no effect on the binding event lifetimes at the lowest trap stiffness used, 0.04 pN·$nm^{-1}$ (Fig. 1c). This suggests that $P_i$ rarely rebound to the active site to induce detachment from the actin filament. However, when the trap stiffness was increased to 0.06 pN.$nm^{-1}$ the mean event lifetimes became much shorter (Fig. 1c), decreasing by 23% compared to 0.04 pN/nm (Fig. 2a). The reduction was most dramatic at the highest stiffness (0.10 pN.$nm^{-1}$), where the lifetimes dropped another 35% compared to 0.06 pN.$nm^{-1}$ (Fig. 2a). Indeed, the binding event lifetimes were significantly higher for the S217A when compared to the WT construct at both 0.04 pN.$nm^{-1}$ and 0.06 pN.$nm^{-1}$, but not significantly at 0.10 pN.$nm^{-1}$ (Fig. 2a). This effect can be seen in more detail by comparing the cumulative distributions and histograms of the WT and S217A (Fig. 2b). The distributions show that S217A construct was less affected by 30 mM $P_i$ at trap stiffnesses of 0.04 and 0.06 pN.$nm^{-1}$. However, at the highest trap stiffness (0.10 pN·$nm^{-1}$) the distributions appear more similar between the WT and S217A constructs. This is consistent with the mutation significantly altering the load-sensitivity of $P_i$-induced detachment.

A silica microsphere held in an optical trap behaves as if attached to a linear spring[38] (Fig. 3a) therefore in addition to measuring the

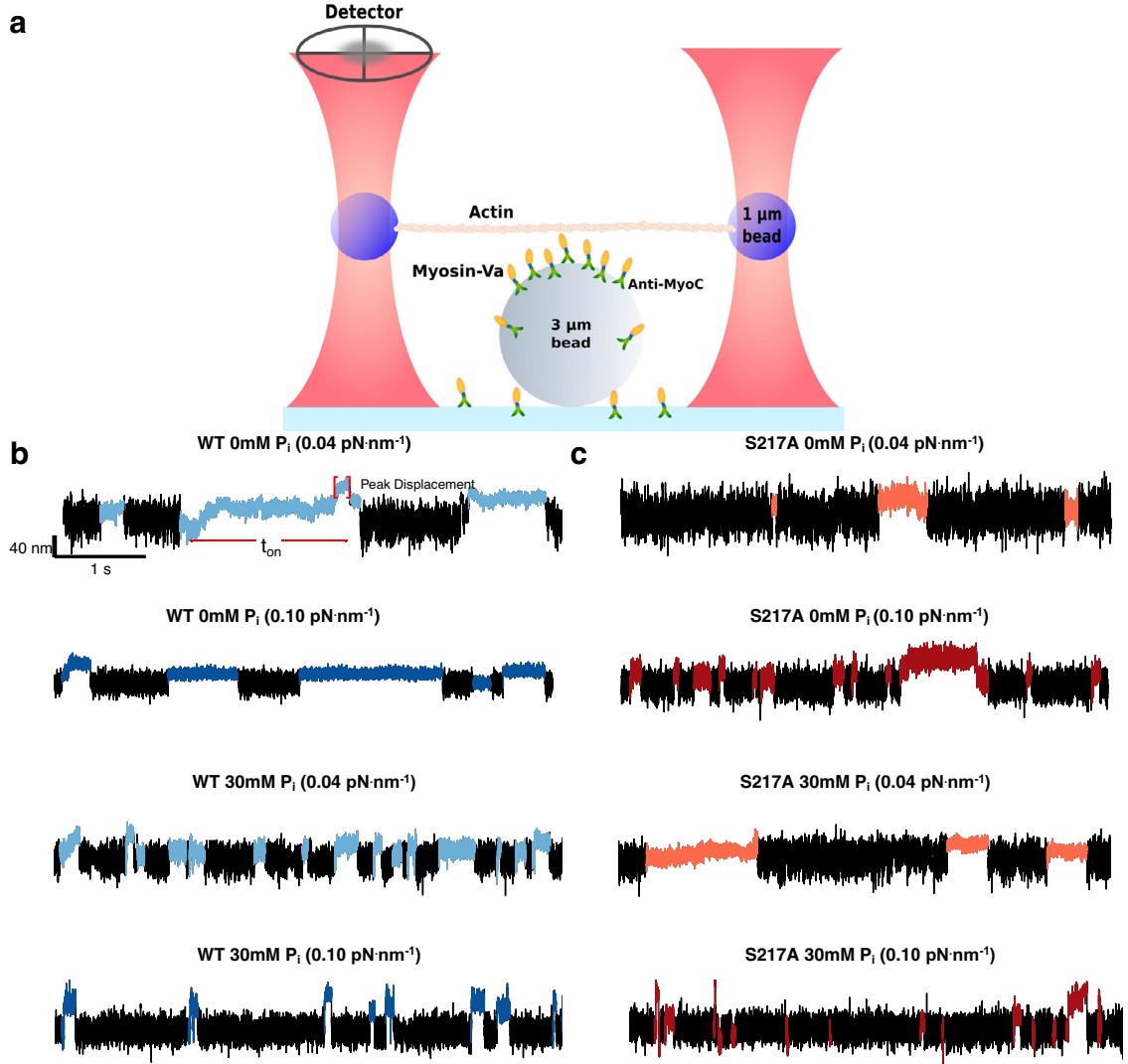

**Fig. 1 | Mini-ensemble laser trap assay and representative displacement records. a** A schematic of the mini-ensemble laser trap assay depicting the geometry of the three-bead laser trap assay. Displacement of the bead-actin-bead assembly was detected using a quadrant photodiode (detector). The myosin Va S1 constructs were adhered to the coverslip surface using c-Myc monoclonal antibody (Thermo-Fisher Scientific Inc., catalog # 13-2500). **b, c** Representative displacement vs. time records for both constructs (WT and S217A) in the absence and presence of 30 mM $P_i$, and at two of the three trap stiffnesses used in the study. Actomyosin binding events were detected using a custom algorithm within the R project for statistical computing (see Methods). The detected events are indicated as the change in color from black to the corresponding color for the laser trap stiffness and [$P_i$]. The duration of the binding event was determined using the same custom algorithm, and the peak force was determined from the peak displacement that was sustained for 5 msec ($t_{on}$), multiplied by the appropriate laser trap stiffness (Force=$k_{trap}$ x displacement from trap center). The scale for the displacement and time near the first trace applies to all records shown.

binding event lifetimes we also measured the peak displacements and forces generated by the mini-ensembles of myosin (Fig. 3d, e). The peak force (F) generated by the ensembles of myosin was determined as the product of trap stiffness ($k_{trap}$) and the distance the trapped microsphere was translocated from the center of the trap (d); i.e. F = $k_{trap}$ * d[39]. Consistent with the well-characterized behavior of an optically trapped microsphere[40,41], the amplitude of the motion caused by Brownian forces was reduced as trap stiffness was increased (Fig. 3b), resulting in a narrowing of the energy well of the laser trap (Fig. 3c). We used this information, and underlying relationships, to determine the effect of $P_i$ on the displacements (Fig. 3d) and forces (Fig. 3e) generated by the ensembles of myosin Va.

Mini-ensembles of WT myosin displaced the actin filament an average of 29 nm in the absence of added $P_i$ (Fig. 3d). The cumulative distribution of forces and displacements are shown in Supplemental Fig. S2. The interactions among the individual myosin heads can be quite complex due to the stochastic nature of attachment and

detachment kinetics, such that some may experience assistive forces while others may experience a resistive load. However, making the assumption that at the peak force all attached heads experience a resistive load we can convert the displacements into a force per head with an estimate of the step size (Table 1). We previously determined that this construct produces a 7 nm powerstroke[33,42], therefore if we assume each head generates a 7 nm step this suggests that, on average, four myosin molecules bound to and displaced the actin filament under these conditions (Table 1). It is interesting to note that this value agrees well with the direct measure of the number of myosin molecules available to bind to the actin filament obtained using STORM super resolution microscopy (Supplementary Figure 1). With the addition of 30 mM $P_i$ the average displacement, and force, was reduced slightly at the lowest trap stiffness (Fig. 3d, e). However, when the trap stiffness was increased in the presence of $P_i$ the displacements (Fig. 3d) and peak forces (Fig. 3e) were significantly greater at 0.06 pN.nm$^{-1}$, and increased further at 0.10 pN.nm$^{-1}$. This translated

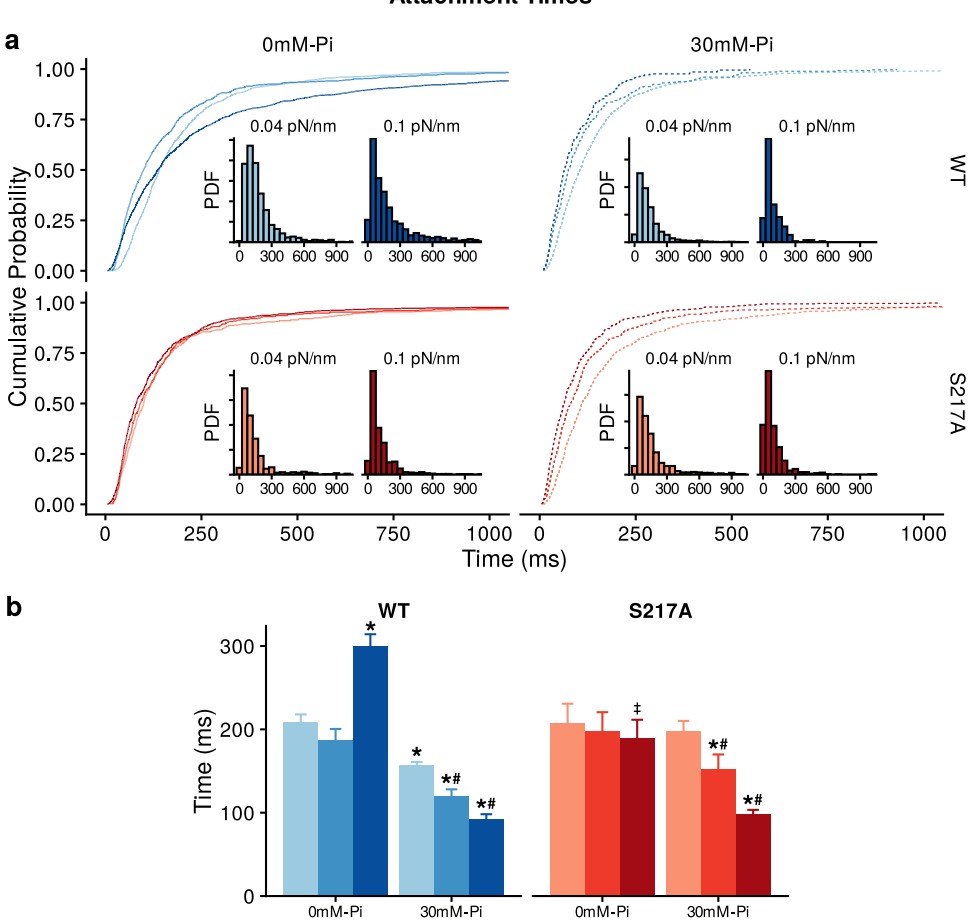

**Fig. 2 | Actomyosin binding event attachment times. a** Main graphs, cumulative distribution of binding event lifetimes for each construct in the presence and absence of $P_i$ as indicated. The three trap stiffness values used (0.04, 0.06 and 0.10 pN.nm$^{-1}$) are represented by increasing darker shades of either blue (WT) or maroon (S217A), darkest shade is highest stiffness. Insets show the same data as a bar graph with the corresponding conditions as indicated. **b** Mean ± SEM of binding event lifetimes for both constructs and each stiffness in the presence and absence of 30 mM $P_i$ as indicated. Significant differences were detected using a non-parametric Kruskal–Wallis ANOVA followed by tests for independent samples using SPSS®. * indicates significantly ($p < 0.05$) different from the control solution (0 mM

$P_i$) for each myosin Va construct. # indicates significantly different from the attached lifetime from WT myosin at the corresponding [$P_i$] and laser trap stiffness. ‡ indicates 0.10 pN.nm$^{-1}$ in the absence of $P_i$ is significantly different from the lifetime at 0.04 pN.nm$^{-1}$ in the absence of $P_i$ for each myosin construct. The number of binding events was 1017, 578 and 1295 for WT in the absence of added $P_i$ for 0.04, 0.06 and 0.10 pN.nm$^{-1}$ respectively and 2322, 276 and 210 in the presence of 30 mM $P_i$ for 0.04, 0.06 and 0.10 pN.nm$^{-1}$. For the S217A construct the number of binding events was 504, 406 and 684 in the absence of added $P_i$ at 0.04, 0.06 and 0.10 pN.nm$^{-1}$ respectively and 584, 416 and 512 in the presence of 30 mM $P_i$ for 0.04, 0.06 and 0.10 pN.nm$^{-1}$.

into higher peak forces at each successive trap stiffness from 0.06 and 0.10 pN.nm$^{-1}$ (Fig. 3e). Interestingly the S217A construct displayed the opposite response to increasing the trap stiffness, with displacements increasing in the presence of $P_i$ at 0.04 pN.nm$^{-1}$, but then progressively decreasing at higher trap stiffnesses (Fig. 3d). As a result of the decreased displacements at higher stiffnesses the peak forces were much lower than those generated by the WT construct (Fig. 3e).

The detachment rates (1/binding event lifetime) were plotted against peak forces generated for both the WT and S217A constructs in Fig. 4a. To quantify the degree of load-sensitivity of $P_i$-induced detachment the data were fit to an equation based on a Bell-bond model[43]:

$$k_1 = k_0 \exp^{\left(\frac{Fd}{kT}\right)} \qquad (1)$$

In the present investigation $k_1$ (i.e. $k_{pi+}$) represents the rate of the $P_i$-induced detachment rate from actin in the presence of resistive load, $k_0$ is the same rate in the absence of load, $F$ is the resistive force, $d$ is the distance to the transition state, which gives a quantitative

measure of load sensitivity of the detachment rate, $k$ is the Boltzmann constant, and $T$ is temperature in Kelvin. This type of analysis has been used previously for quantifying the load sensitivity of other kinetic steps in myosin's cross-bridge cycle[35,44].

This analysis predicted that $P_i$ would rebind to WT myosin Va at a rate of 5 s$^{-1}$ in the absence of a resistive load, but that rate increased to 10 s$^{-1}$ in the presence of a 4pN load (Fig. 4a). The curvature of this relationship, defined by the parameter $d$ (units of nm), is indicative of the degree of load-dependence of the reaction rate, which for WT myosin was 0.6 nm. Performing this same analysis on the S217A construct revealed that it had a much lower rate of $P_i$ rebinding in the absence of load (1.4 s$^{-1}$), but displayed a much greater sensitivity to the increasing resistive load, resulting in $d$-value that was almost seven-fold higher than WT myosin (4.1 nm). Indeed, a statistical analysis of the fit parameters revealed that both $k_0$ and $d$ were significantly different for S217A, compared to WT myosin (Fig. 4b).

## Discussion

We characterized the load-dependence of $P_i$-induced detachment of a mini-ensemble of myosin molecules in a three-bead laser trap assay

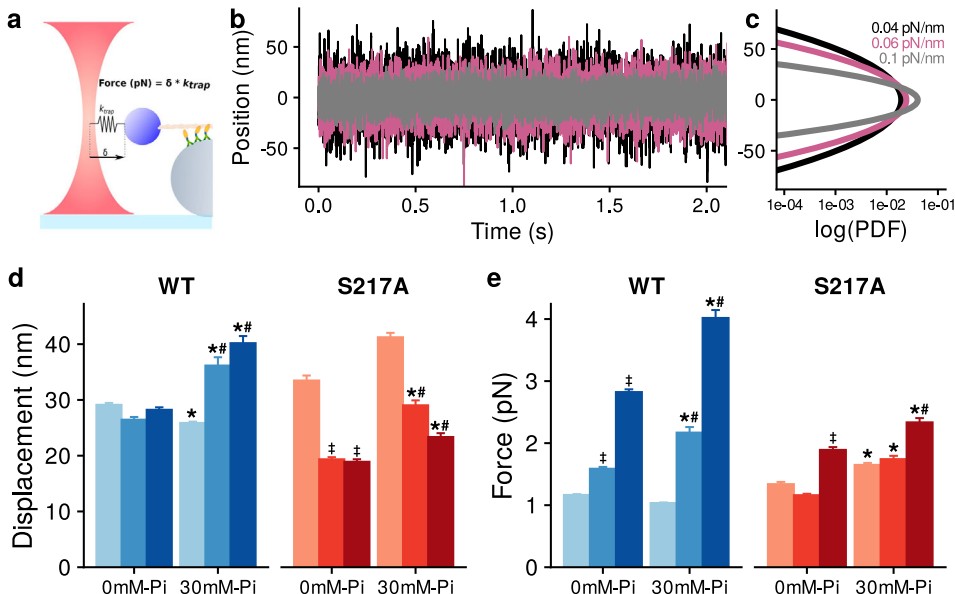

**Fig. 3 | Binding event displacements and forces. a** A schematic of the laser trap assay. Force was equal to trap stiffness ($k_{trap}$) multiplied by the displacement from trap center ($\delta$) **b** Raw displacement records at increasing trap stiffness values from 0.04 (black) to 0.06 (pink) to 0.10 pN.nm$^{-1}$ (gray). **c** The power density function (PDF) at each trap stiffness from 0.04 to 0.10 pN.nm$^{-1}$, determined from the raw displacement records of bead-actin-bead assembly alone (Fig. 3b). Significant differences were detected using a non-parametric Kruskal-Wallis ANOVA followed by two-tailed tests for independent samples using SPSS®. No adjustment was made for multiple comparisons **d** Mean ± SEM for peak displacements at each trap stiffness, in the absence and presence of 30 mM $P_i$. * indicates significantly ($p < 0.05$) different from the control, 0 mM $P_i$. # indicates significantly different from the

attached lifetime for WT myosin at the corresponding [$P_i$] and laser trap stiffness. ‡ indicates that 0.10 pN.nm$^{-1}$ in the absence of $P_i$ is significantly different from the lifetime at 0.04 pN.nm$^{-1}$ in the absence of $P_i$ for each myosin construct.
**e** Mean ± SEM of the peak force generated at each stiffness and [$P_i$]. *, # and ‡ same as for Fig. 3d. The number of binding events was 1017, 578 and 1295 for WT in the absence of added $P_i$ for 0.04, 0.06 and 0.10 pN.nm$^{-1}$ respectively and 2322, 276 and 210 in the presence of 30 mM $P_i$ for 0.04, 0.06 and 0.10 pN.nm$^{-1}$. For the S217A construct the number of binding events was 504, 406 and 684 in the absence of added $P_i$ at 0.04, 0.06 and 0.10 pN.nm$^{-1}$ respectively and 584, 416 and 512 in the presence of 30 mM $P_i$ for 0.04, 0.06 and 0.10 pN.nm$^{-1}$.

(Fig. 1). By varying the stiffness of the laser trap, the resistive force on the myosins was progressively increased (Fig. 3). The increase in force decreased binding event lifetimes by as much as 50% in the presence of elevated $P_i$ (Fig. 2). We used an equation based on a Bell-bond model[43,44] to provide a quantitative assessment of the load-dependence of the rate of $P_i$ rebinding to myosin's active site (Fig. 4a). We also found that a mutation in the switch I region of myosin's active site (S217A) made the myosin Va seven-fold more sensitive to a restive load (Fig. 4a). This suggests that this region may be involved in mediating the load-dependence of $P_i$-rebinding to the active site. $P_i$-release and rebinding kinetics are putatively coupled to the powerstroke[22] therefore these findings may offer insights into the mechanism of energy transduction by this prototypical molecular motor.

This is the first characterization of the load-dependence of $P_i$-rebinding using a mini-ensemble of myosin molecules. We found that as the myosins experienced the higher load of a stiffer laser trap the rate of detachment from actin significantly increased in the presence of 30 mM $P_i$. This is consistent with $P_i$ rebinding to actomyosin in an AM.ADP state and accelerating its detachment from an AM.ADP.$P_i$ to a M.ADP.$P_i$ state (Fig. 4c). Thus, in the presence of elevated $P_i$, and a resistive load, most binding events were likely ended by $P_i$-induced detachment rather than ATP-induced detachment from rigor (Fig. 4c).

Resistive load putatively slows the rate of ADP-release from the active site[36,44,45], and the prolongation of the event lifetime at the highest stiffness in the absence of $P_i$ (Fig. 2b) is consistent with prior observations[36]. Therefore, as the resistive load increased the myosins may have become more vulnerable to $P_i$ rebinding, because the duration of the AM.ADP state was prolonged by load. This explanation can be parsed even further, as the AM.ADP state is thought to be comprised of pre- and post-isomerization states[25], with only the pre-

isomerization state thought to be vulnerable to the rebinding of $P_i$[26,27,29,37]. Therefore, it is likely that load slows the rate of the AM.ADP isomerization, and by doing so increases actomyosin's affinity for $P_i$.

Interestingly, as the stiffness of the traps was increased the magnitude of the displacements, and forces, increased slightly (Fig. 3d, e). A simple explanation for this observation may be that the increasing stiffness of the traps limited their motion in the z-axis, making it easier for the myosin molecules to access the actin filament. However, assuming each myosin Va molecule generated a 7 nm displacement[33,42] and only interacted once with the actin filament during an event, the displacement data suggest that only one additional myosin, or even less, bound to actin at the higher stiffnesses (Fig. 3d). Thus, this explanation seems unlikely to explain the much larger forces reached at the higher stiffnesses (Fig. 3d). Indeed, a 7 nm displacement per myosin molecule translates to 0.28 pN per myosin head at the lowest stiffness (0.04 pN.nm$^{-1}$) and 0.70 pN per myosin head at the highest trap stiffness (Table 1). Thus, despite an additional head binding to the actin filament the force per myosin head was still higher at the higher trap stiffnesses. This suggests that as the stiffness was increased each head generated more force. It is this higher load per head that likely drove the load-dependent decrease in the lifetime of the binding events (Fig. 2).

To provide a quantitative estimate of the load-dependence of the rate of $P_i$-induced detachment the force vs attachment time (Fig. 4a) these data were fit with an equation based on a Bell-bond model[43]. In this equation the d-value characterizes the curvature of this relationship, and putatively corresponds to the distance to the transition state[44,46]. The value of 0.6 nm is consistent with the rate being sensitive to the resistive load. It also predicts the rate in the absence of load (y-int), which at ~5 s$^{-1}$ is consistent with the value previously reported for the very low loads generated in a single molecule three-bead assay,

**Table 1 | Values displayed are mean ± SEM of peak forces and displacements**

| Myosin | $P_i$ (mM) | Stiffness (pN.nm⁻¹) | Displacement (nm) | Force (pN) | Heads | Force/Head (pN) |
|---|---|---|---|---|---|---|
| WT | 0 | 0.04 | 29.1 ± 0.3 | 1.17 ± 0.01 | 4.2 | 0.28 |
| WT | 0 | 0.06 | 26.5 ± 0.5 | 1.59 ± 0.03‡ | 3.8 | 0.42 |
| WT | 0 | 0.10 | 28.2 ± 0.4 | 2.82 ± 0.04‡ | 4.0 | 0.71 |
| WT | 30 | 0.04 | 25.9 ± 0.2* | 1.03 ± 0.01 | 3.7 | 0.28 |
| WT | 30 | 0.06 | 36.2 ± 1.5*# | 2.17 ± 0.09*# | 5.2 | 0.42 |
| WT | 30 | 0.10 | 40.2 ± 1.3*# | 4.02 ± 0.13*# | 5.7 | 0.70 |
| S217A | 0 | 0.04 | 33.5 ± 0.9 | 1.34 ± 0.04 | 4.8 | 0.28 |
| S217A | 0 | 0.06 | 19.4 ± 0.4‡ | 1.16 ± 0.02 | 2.8 | 0.41 |
| S217A | 0 | 0.10 | 18.9 ± 0.4‡ | 1.89 ± 0.04‡ | 2.7 | 0.70 |
| S217A | 30 | 0.04 | 41.2 ± 0.8 | 1.65 ± 0.03* | 5.9 | 0.28 |
| S217A | 30 | 0.06 | 29 ± 0.9*# | 1.74 ± 0.05* | 4.1 | 0.42 |
| S217A | 30 | 0.10 | 23.4 ± 0.7*# | 2.34 ± 0.07*# | 3.3 | 0.71 |

The average number of myosin heads bound to actin at the displacement was determined by dividing the total displacement by the size of the powerstroke (7 nm), as reported for the same myosin construct in a single molecule laser trap assay[42]. The peak force was calculated by dividing this number by the number of myosin molecules to determine the force per myosin head. The number of binding events was 1017, 578 and 1295 for WT in the absence of added $P_i$ for 0.04, 0.06 and 0.10 pN.nm⁻¹ respectively and 2322, 276 and 210 in the presence of 30 mM $P_i$ for 0.04, 0.06 and 0.10 pN.nm⁻¹. For the S217A construct the number of binding events was 504, 406 and 684 in the absence of added $P_i$ at 0.04, 0.06 and 0.10 pN.nm⁻¹ respectively and 584, 416 and 512 in the presence of 30 mM $P_i$ for 0.04, 0.06 and 0.10 pN.nm⁻¹. Significant differences were detected using a non-parametric Kruskal-Wallis ANOVA followed by two-tailed tests for independent samples using SPSS®. No adjustments were made for multiple comparisons. * indicates significantly ($p < 0.05$) different from the control, 0 mM $P_i$. # indicates significantly different from the attached lifetime from WT myosin at the corresponding [$P_i$] and laser trap stiffness. ‡ indicates 0.10 pN.nm⁻¹ in the absence of $P_i$ is significantly different from the lifetime at 0.04 pN.nm⁻¹ in the absence of $P_i$ for each myosin construct.

using the same construct and [$P_i$][42]. In transition state theory, force is thought to tilt the energy landscape[46], which in the present study would alter the height of the well for the AM.ADP state making easier for $P_i$ to rebind and induce detachment from the actin filament. Figure 4c is drawn to represent this interpretation using a 2-D representation of the energy landscape.

The load-dependence of $P_i$-induced detachment observed in the present study with WT myosin Va is consistent with prior observations at the single molecule level using a similar construct[35]. As part of a larger study focused on visualizing reversals of the powerstroke, Sellers and Veigel[35] observed that the detachment rate in the presence of 10 mM $P_i$ was accelerated by the application of a resistive load. Their observed $d$ value (0.6 nm) is remarkably consistent with the 0.6 nm value observed in the present study (Fig. 4a). This is especially striking given the use of higher resistive loads (-1-6 pN per head) and a lower $P_i$ concentration (10 mM). Therefore, this suggests that myosin may have a force threshold above which its affinity for $P_i$-rebinding is saturated, similar to the effect $P_i$ has on maximal isometric force in muscle[47].

The data from the WT myosin Va construct in the absence of added $P_i$ (Fig. 2) are also consistent with prior observations using a two-headed construct of myosin Va[37]. In the absence of $P_i$ they observed that the two-headed myosin processed against the spring-like load of the laser trap before pausing for long durations at stall force. By comparison, in the presence of 40 mM $P_i$, the myosin displaced the actin filament by the same distance, reaching the same stall force, but the duration of time spent at the stall force was dramatically reduced. These data suggest that $P_i$ accelerated detachment from the AM.ADP.$P_i$ state, consistent with the detachment pathway we favor based on the present data, using the single-headed construct (Fig. 4c). Our findings are also consistent with observations in muscle fibers (myosin II) where the effect of $P_i$ on the force-velocity relationship is dependent on the load, with $P_i$ having little effect on unloaded

shortening velocity but causing dramatic reductions in maximal isometric force[48]. Specifically, elevated levels of $P_i$ have little effect on unloaded shortening velocity in a muscle fiber[23,48,49], or on actin filament velocity in an in vitro motility assay (at mM ATP and neutral pH)[50–52], because in the absence of a resistive load $P_i$-release is virtually irreversible. Findings such as these have led to the well-established idea that resistive loads prolong the lifetime of the state from which $P_i$-release is readily reversible[8,22,53], thus rebinding and subsequent detachment from actin can readily occur. For myosin Va this induces dissociation from the actin filament under load[37], and in a muscle fiber this is evidenced as a $P_i$-dependent reduction in isometric force[26,27]. Thus, while the kinetics may differ, the load sensitivity of $P_i$-rebinding to the nucleotide-binding site may be conserved across several members of the myosin superfamily.

To begin to identify the structures and regions of myosin responsible for mediating this load-sensitivity we repeated the experiments using a construct with a S217A mutation. This residue was chosen because it lies in a key region of myosin's nucleotide binding site (switch I) that it is thought to play a critical role in the release of $P_i$ from the active site[31,54]. This construct's response to the increased trap stiffness was strikingly different from that of WT (Figs. 1 & 2). Firstly, in the absence of $P_i$ the increased stiffness did not prolong the duration of event lifetimes, in contrast to the prolongation observed in the WT (Fig. 2), suggesting the rate of ADP-release was load insensitive. Likewise, in the presence of $P_i$ at the lowest trap stiffness (0.04 pN·nm⁻¹) the attachment lifetime was largely unaffected by $P_i$ (Fig. 2a), suggesting that $P_i$ did not rebind to the active site and accelerate detachment as it did in the WT construct. This is consistent with prior observations at the single molecule level[42] and suggests that in the present study, at this low trap stiffness, the binding events were terminated not by $P_i$ but by ATP. However, at the higher trap stiffnesses the reductions in the attachment lifetime were greater than in WT (Fig. 2a). This distinctly different response to load caused a dramatic change in the quantitative assessment of the load-dependence. Indeed, the $d$-value observed in the Bell-bond equation suggested that it was seven-fold more sensitive to a resistive load than the WT construct (Fig. 4b, c). The dramatic difference in the $d$-value suggests that this region of the nucleotide binding site may be involved in mediating the load-dependence of $P_i$-rebinding to the nucleotide-binding site.

Presumably, $P_i$ rebinds to the active site through the same pathway from which it was released. There are at least two potential exit routes[12,55]; the first is through an opening created by an actin-induced conformational change in the switch II element of the active site[21,30]. In the absence of load this opening is thought to be a transient one, quickly closing down once $P_i$ exits the active site[8,12]. However, a resistive load is thought to prolong the lifetime of this open-state[12,21,22], which may create a pathway for $P_i$ to rebind to the active site, and once there it induces the opening of the actin-binding cleft which leads to the acceleration of detachment from the AM.ADP.$P_i$ state. However, it is unclear how a mutation in switch I might alter the load-dependence of rebinding through switch II, unless it has an effect that propagates from residue 217 in switch I to the nearby switch II element.

The second proposed exit for $P_i$ is through a gap between the P-loop and switch I[55,56]. This pathway is also thought to be load sensitive[55] and thus it is possible that loss of the OH group at this site, with the serine to alanine substitution at 217, affects the role that this residue plays in interacting with $P_i$ as it exits[31], and re-enters, the nucleotide binding site. This seems the more likely explanation given that the substitution is in switch I. It is also possible that rebinding occurs through a different pathway than $P_i$ release, as it has been suggested that there are six possible exit routes, based on studies utilizing molecular dynamics simulations[56]. Regardless of the specific conformational changes our findings suggest that this region of the active site contains elements that are sensitive to a resistive load, and in response to that load they modulate the kinetics of $P_i$ rebinding, and

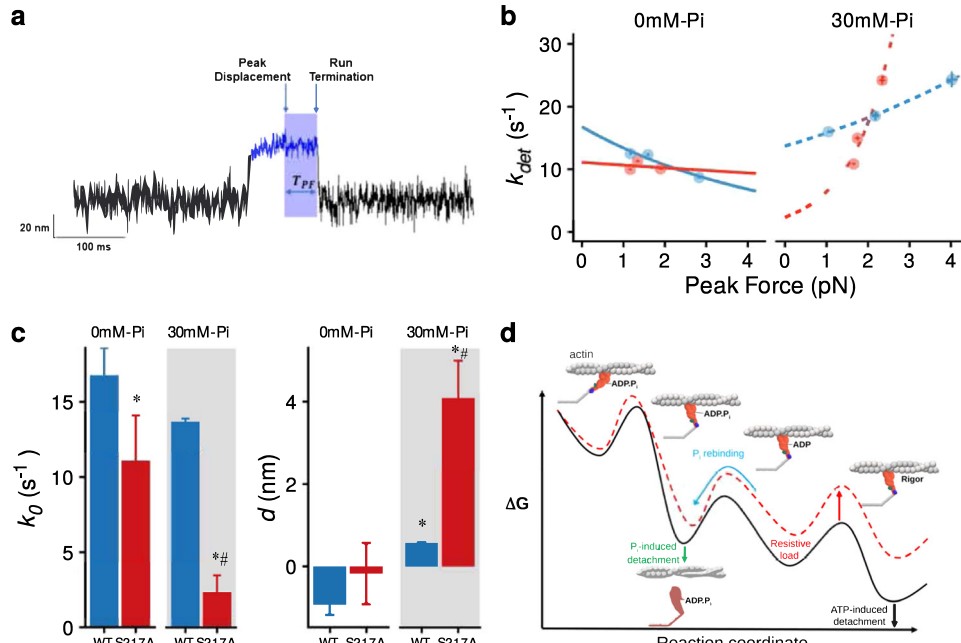

**Fig. 4 | Load-dependence of the $P_i$-induced detachment rate and the hypothesized effect on the energy landscape. a** A raw displacement record depicting the method used to determine the time spent at peak force ($T_{PF}$), identified by sliding a 5 ms window over the data. Once the peak was determined the time spent at that peak force was determined until the run was terminated (shaded area), see Methods for more detail. **b** The durations at peak force were converted to rates ($k_{det}$) and plotted against the peak force. Values represent the mean ± SEM. Data were fit with the Bell-bond equation: $k_{det} = k_o \times exp(Fd/kT)$. The fit parameters for WT were $k_o = 16.8 \pm 1.8$ s$^{-1}$ (mean ± SEM) and the $d$ value was $-0.9 \pm 0.3$ nm (mean ± SEM) in the absence of $P_i$ and 13.7 ± 0.2 s$^{-1}$ for $k_o$ and 0.6 nm for $d$ in the presence of 30 mM $P_i$. For the S217A construct $k_o$ was 11.1 ± 0.3.0 s$^{-1}$ and a $d$-value of $-0.2 \pm 0.7$ nm in the absence of $P_i$, and $k_o$ was 2.5 ± 1.1 s$^{-1}$ and the $d$-value 4.1 ± 0.9 nm in the presence of $P_i$. **c** Bar graph of means ± SEM of the fit parameters to the Bell-bond equation. Differences between the fit parameters to the Bell-bond equation were determined by constructing 95% confidence intervals around the point

estimates. * indicates significantly different from WT at 0 mM $P_i$, The number of binding events was 1017, 578 and 1295 for WT in the absence of added $P_i$ for 0.04, 0.06 and 0.10 pN.nm$^{-1}$ respectively and 2322, 276 and 210 in the presence of 30 mM $P_i$ for 0.04, 0.06 and 0.10 pN.nm$^{-1}$. For the S217A construct the number of binding events was 504, 406 and 684 in the absence of added $P_i$ at 0.04, 0.06 and 0.10 pN.nm$^{-1}$ respectively and 584, 416 and 512 in the presence of 30 mM $P_i$ for 0.04, 0.06 and 0.10 pN.nm$^{-1}$. **d** A speculative energy landscape depicts the effect of load on the $P_i$-induced detachment rate. There is a large drop in free energy associated with the generation of the powerstroke, suggesting it is not readily reversed[10]. Increasing resistive load tilts the landscape such that it is easier for $P_i$ to rebind to the AM.ADP state allowing it to reform a post-powerstroke, AM.ADP.$P_i$ state from which detachment is rapid. This resistive load makes it less likely that myosin will complete ADP-release and ATP-induced detachment. The red dotted line depicts the effect that a higher resistive load (i.e. greater trap stiffness) has on the energy landscape.

likely release. This represents an important advance in our understanding of the nature of coupling between force generation and the release of the products of ATP-hydrolysis and thus how the chemical and mechanical events are coupled.

It is noteworthy that in the presence of $P_i$ the increased laser trap stiffness did not decrease the magnitude of the displacements generated, rather they were increased in the WT construct (Fig. 3a), despite dramatically reducing the duration of attachment (Fig. 2). This observation seems difficult to reconcile with a model in which $P_i$-release precedes the powerstroke, because this model[8,12,21] requires $P_i$ to leave the active site before the powerstroke can occur. Therefore, the rebinding of $P_i$ to the active site should prevent the powerstroke and thus prevent or inhibit displacements as load is increased, because $P_i$ will more readily rebind to the active site at higher resistive loads. Thus, in the present study the displacements should have decreased as the resistive force became larger to be consistent with this type of model, however we observed the opposite effect with displacements increasing for the WT construct at increasing resistive forces (Fig. 3d). Displacements decreased for the S217A construct, however this is likely due to a compromised ability to transition from the weak-to-strongly bound state[33]. Thus, as the load increased additional myosin molecules were less likely to bind to and translocate the actin filament because they could not form a strong bond with actin, consistent with the load dependence of weak-binding[57].

A potential alternative mechanism, which could be consistent with $P_i$-release occurring prior to the powerstroke, is if $P_i$ rebound not

to the active site, but to an alternative or secondary site[58]. In this case the $P_i$ would have rebound to a secondary binding site[58] either within the putative $P_i$-release tunnel[21] or at another distal location on the surface of myosin[58]. However, any such mechanism would have to accelerate myosin's detachment from actin to be consistent with the decreased event lifetime (Fig. 2a). Thus, this secondary $P_i$-binding site would have to have allosteric control over myosin's affinity for actin, i.e. the opening of actin-binding cleft. While this is possible we know of no evidence to support the idea that $P_i$ has such an allosteric effect on actin-binding cleft dynamics through a binding site outside of the active site. Additionally, there is a large body of literature suggesting that under resistive load $P_i$ readily rebinds to the active site[14,26,29,47,59]. Therefore our WT data seem most consistent with a model in which the powerstroke precedes the release of $P_i$ from the active site[11,13,14,26,33]. Thus, the most likely scenario that explains the WT data is that myosin went through a powerstroke upon formation of a strong-bond with actin, and subsequently released $P_i$. A new $P_i$ then rebound to acto-myosin in a post-powerstroke, AM.ADP state, transiently creating an AM.ADP.$P_i$ state that rapidly dissociated from actin to become an M.ADP.$P_i$ state (Fig. 4c)[50,53]. Dissociation may occur from a post-powerstroke state[50,53], or after a reversal of the powerstroke[14,26,35]. The idea that the powerstroke precedes $P_i$ release is consistent with the rapid rate (1000–5000/s) of myosin's powerstroke, directly observed in an ultra-fast laser trap assay[14,57]. This rate is at least an order of magnitude faster than actin-stimulated $P_i$-release[33]. An interesting implication of a powerstroke-first model is that it suggests that the

driving force of myosin's powerstroke is not the release of $P_i$ from the active site, but the formation of the strongly-bound state between actomyosin, an idea that has been proposed previously[60].

In conclusion, the rate of Pi-induced detachment of a mini-ensemble of myosin molecules from actin is load-dependent, displaying a $d$-value of 0.6 nm (Fig. 4b, c). This finding is consistent with resistive load accelerating the rebinding of $P_i$ to myosin's active site and inducing detachment from an AM.ADP.$P_i$ state. Interestingly, the magnitude of the displacements generated in the presence of elevated $P_i$ increased as the resistive load increased. This is most consistent with a model in which the powerstroke occurs prior to the release of $P_i$ from the nucleotide binding site. We probed the structural basis of this load-sensitivity and found that a substitution (S217A) in the switch I element of the nucleotide binding site dramatically altered the load-dependence of this rate increasing the $d$-value to 4.1 nm. This suggests that this region of the nucleotide-binding region may be involved in mediating myosin's load-sensitivity to the rebinding of $P_i$. Thus, these findings provide insights into the mechanism of energy transduction by myosin Va and suggest a region of the active site that may govern the load-sensitivity of this reaction.

## Methods

### Protein construction, expression, and purification

Expressed, single-headed myosin Va S1 constructs were prepared from a chicken myosin Va sequence (residues 1-792), as previously described[33]. Briefly, the expressed version contained only the first IQ domain, with an N-terminal tetracysteine motif, and a C-terminal Myc and FLAG tags, that were used for purification and for aiding attachment to the coverslip surface. The S217A construct was created using Quikchange site directed mutagenesis (Stratagene) and co-expressed with calmodulin in baculovirus. Both constructs were purified using FLAG affinity chromatography. Purified proteins were frozen in liquid nitrogen with 5% sucrose in 20 uL aliquots, that were then thawed on the day of an experiment. Three (WT) to five (S217A) different protein preparations were sampled to obtain the present data. Actin was isolated from chicken skeletal muscle by first preparing acetone powder and then isolating it as g-actin in low salt dialysis before repolymerizing in the presence of 1 mM $MgCl_2$, 1 mM ATP and 60 mM KCl. The laser trapping assays used stabilized actin filaments labeled with phalloidin/TRITC and phalloidin/biotin in a 50/50 mixture.

### Mini-ensemble laser trap assay

A three-bead laser trap assay (Elliot Scientific Inc., Model E3200) was used to determine the duration of strong binding to an actin filament and the magnitude of force generated by mini-ensembles of single-headed myosin Va, as previously detailed[53] with minor modifications described below. Myosin was adhered to a nitrocellulose-coated microscope slide where 3 μm silica microspheres served as pedestals for myosin to attach. The flow-cell was precoated with c-Myc monoclonal antibody (Thermo-Fisher Scientific Inc., catalog # 13-2500) to serve as a substrate for the myosin Va S1 to bind to. The 200 ug powder was brought up in 100 uL of $ddH_2O$ and diluted 1:100 before flowing into the flowcell. The myosin was then added at a concentration of 10 μg/mL. The flowcell was then incubated for 5 min with bovine serum albumin (0.5 mg/mL) to inhibit non-specific interactions between the actin filament and the nitrocellulose-costed coverslip surface. The experimental buffer contained; 1 μm streptavidin-coated silica microspheres (Bangs Labs Inc.), 50:50 biotin/TRITC-labeled actin, an oxygen scavenger system (29 mM glucose, 1.5 mM glucose oxidase, and 80 units catalase), within a low-salt buffer (91 mM KCl, 1 mM EGTA, 4 mM $MgCl_2$, and 1 mM DTT, 100 μM ATP, at 25 °C and a pH of 7.0). The concentration of KCl was reduced for experiments with 30 mM $P_i$ to maintain a constant total ionic strength of 125 mM.

A three-axis piezo-controlled stage (Mad City Labs, Inc.) was maneuvered to attach a biotin/TRITC-labeled actin filament to two 1 μm neutravidin-coated silica microspheres (Bangs Labs, Inc.) held in two-time shared optical traps. Once an actin filament was attached to the 1 μm microspheres, the filament was extended between the optically trapped microspheres, to a apply pretension of 3-4pN to the actin filament. The bead-actin-bead dumbbell assembly was then lowered into close proximity of the 3 μm myosin-coated pedestal bead. The displacements generated by actomyosin binding activity were tracked from the output from quadrant photodiode via interferometry at a sampling rate of 5 kHz. To change the resistive load experienced by the myosin the trap stiffness was increased by increasing the intensity (i.e. power) of the laser beam used to establish the optical traps. The three values of two trap stiffness used were 0.04, 0.06 and 0.10 pN.nm$^{-1}$.

A total of 8804 actomyosin binding events were detected and recorded, from sampling 209 different myosin-coated 3 μm pedestals. Two-trap stiffness (the sum of the stiffness of both traps), the value that the myosin works against[41], was determined at each laser power using the equipartition theorem[61]. This stiffness value was then multiplied by the peak displacement (see Fig. 1a) to estimate the highest resistive force experienced by the ensembles of myosin during a binding event. Binding event durations were converted to rates ($1/t_{on}$) and plotted against the estimated forces (Fig. 4b). These data were then fit with an equation based on a Bell-bond model[44], Eq. (1).

### Data processing and analyses

Recorded data was analyzed using custom programs written within The R Project for Statistical Computing (v4.04, R Core Team). These programs are based on a two-state Hidden-Markov Model[62] which was used for binding event identification using running mean and variance transformations of the raw displacement data from the laser trap assay. The windows used to calculate mean and variance had a width of 100 datapoints and progressed with one-quarter overlap over the entire displacement record. The start and end estimations of each binding event were inferred from the original raw data using the output of the Hidden-Markov Model analysis, multiplying the first and last running window position (or indexed data) by the running window width. A changepoint analysis[63] was then applied to a small subset of data surrounding the transition points (2 window widths on either side of the start and end of each binding event). The changepoint analysis was performed using a running mean and variance of the displacement data (50 data point window progressing one data point at a time) to identify the most probable datapoint of the transition into or out of a binding event. The peak displacements in ensemble runs were determined using a 5 ms running mean window that identified the window with the greatest displacement, within the two changepoint-identified transitions (start and end points) of actomyosin binding events, and returned that window's value as peak displacement (see Fig. 1a). The time spent at peak force ($T_{PF}$) was determined to calculate changes in the detachment rate as a function of load. The starting point of $T_{PF}$ began at the greatest displacement window identified within the two changepoint-identified transitions, as described above, and ended at the changepoint-identified end point of the actomyosin binding event (i.e. run termination). The total time between these two points was then calculated for all actomyosin binding events and used for the calculation of $k_{det}$.

### Statistical analyses

The detected events were analyzed using SPSS® (version 28.0.1.0). The binding event lifetimes were found to be non-normally distributed based on a Kolmogorov-Smirnov test of Normality, therefore we used a non-parametric approach to detecting significant differences ($p < 0.05$). A Kruskal-Wallis ANOVA for independent samples was used to look for, and locate, differences among the

different conditions and constructs. The displacements and forces generated by the mini-ensembles of myosin were also non-normally distributed based on the same test of Normality, therefore a Kruskal–Wallis ANOVA was also used to assess and locate any significant differences. Differences between the fit parameters to the Bell-bond equation were determined by constructing 95% confidence intervals around the point estimates.

## Stochastic optical reconstruction super resolution microscopy (STORM)

To determine the number of myosin Va molecules within reach of the actin filament during the mini-ensemble laser trap assay (Fig. 1) a parallel set of experiments were performed using STORM (See Fig S1)[34]. Myosin Va was adhered to a nitrocellulose coated flow cell at the same concentration and using the same attachment strategy, as in the mini-ensemble laser trap assay. The myosin molecules were then labeled by introducing TRITC-labeled actin filaments that had been extensively sonicated and vortexed to fragment them. This was done within a buffer without ATP so that they bound to myosin and formed a rigor attachment. The buffer also included the oxygen scavenging system described above to maintain the activity of fluorophores constant during the duration of data collection. The sample was then imaged using a STORM super-resolution microscope system, which is built around a Nikon TiE microscope with a Hamamatsu Flash 4.0 sCMOS camera, a piezo-controlled stage (Physik Instruments, Inc.), a LUN-V laser unit (Nikon Instruments Inc.) and N-STORM 4.0 hardware.

The fragmented, TRITC-labeled actin filaments were excited with a 561 nm laser in TIRF illumination mode and the fluorophore blinking behavior was acquired for 60,000 frames at a rate of 100 Hz. Each point within the reconstructed images was fit with a Gaussian point spread function to determine the center of intensity. These were then used to generate the 2-D (x,y) reconstructed images of the actin fragments on the myosin-coated surface. These points were then used to determine the average distance between any two molecules (Fig. S1C) using the nearest-neighbor analysis tool within Nikon NIS-Elements software within the GA3 module. These data were plotted as a histogram and fit with a Gaussian curve to determine the center of the distribution, which was 119 nm. To estimate the number of heads within reach of the actin filament we determined the length that would be within 20 mn (the height of a single myosin head) of the 3um pedestal bead, assuming the center of the filament was in contact with the pedestal surface (See Supplemental Material Fig. S1). This approach suggested that 4-5 myosin heads were capable of interacting with the actin filament (Fig. 1d), which is consistent with maximum displacements observed in the mini-ensemble laser trap assay (Fig. 3d).

### Reporting summary

Further information on research design is available in the Nature Portfolio Reporting Summary linked to this article.

## Data availability

Source data are provided with this paper. The datasets generated and/or analyzed during the current study are available from the corresponding author upon reasonable request. The Source data underlying Fig. 1b are provided as Source Data 1. The Source data underlying Fig. 2a, b are provided as Source Data 2. The Source da`ta underlying Fig. 3b–f are provided as Source Data 3. The Source data underlying Fig. 4 are provided as Source Data 4. Source data are provided with this paper.

## Code availability

The custom software and underlying code used to analyze the data is available as an R-package (https://github.com/brentscott93/lasertrapr).

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

## Acknowledgements

This work was supported by grants from the NIH (R01GM135923-01) and the American Heart Association (18IPA34170048) to E.P.D. The STORM microscopy data was gathered in the Light Microscopy Facility and Nikon Center of Excellence at the Institute for Applied Life Sciences, UMass Amherst with support from the Massachusetts Life Sciences Center. We also thank Katherine Boutin for assistance collecting STORM data. E.P.D. thanks Joseph and Catherine Debold for a lifetime of support.

## Author contributions

E.P.D., C.M.Y., B.S., and C.M. designed the experiments and edited the manuscript. C.M. collected and analyzed the data. B.S. helped prepare figures. L.K.G. expressed and purified the myosin Va constructs. J.C. assisted with the STORM data collection and analysis of the images. E.P.D. wrote the manuscript.

## Competing interests

The authors declare no competing interests
