## [Peer Review File · Nature Communications]

nature portfolio

Peer Review FileEditorial Note: Parts of this Peer Review File have been redacted as indicated to remove third-party material where no permission to publish could be obtained.

REVIEWER COMMENTS

Reviewer #1 (Remarks to the Author):

This paper by Marang et al. presents results about the load dependence of Pi-release from the myosin active site and its relation to force generation. A three-bead optical tweezers system is used to probe the generation of force and displacements produced by wild-type myosin V motor domains and myosin V motor domains with an S217A mutation in switch I, a key nucleotide and Pi-binding loop in myosin. The studies are performed at varied concentration of inorganic phosphate, Pi, focusing on 30 mM Pi.

Specifically, the paper provides results on the load dependence of [Pi]-effects on force-generation by mini-ensembles of myosin V in the presence and absence of the mutation S217A. The authors find that external load increases the rate of actomyosin detachment due to Pi-binding. Moreover, based on data suggesting that the load-dependence of the detachment rate is increased in the presence of the S217A mutation the authors conclude that switch I is important for the force-dependent nature of Pi-induced detachment.

The rationale of the study is clearly laid out in the Introduction. The Methods seem to be described in sufficient detail to repeat the experiments and the experiments appear to be carefully executed. The paper is also generally well written except for some minor issues raised below. However, I have some major concerns with the paper in other regards, particularly the extent to which the experimental data support the major conclusions:

1. This is one in a series of studies by the present authors and others of the effects of the S217A mutation in myosin V (1-3) and corresponding mutations in other myosins (4). The novelties presented here, compared to the previous studies, seem somewhat limited. This is for instance suggested by comparison to the recent paper by Scott and co-workers (3) who used optical tweezers to study motor function of myosin V in the presence of the S217A mutation and at varied Pi. What is new here compared to previous work would seem to be the increased sensitivity to external load of Pi-induced cross-bridge detachment in the presence of the S217A mutation. However, the load dependence of this detachment rate and its Pi dependence are known previously for wild type myosin (5, 6 7, 8). Whereas the increased load-sensitivity with the S217A mutation is an interesting finding, experimental complexities (summarized in the next point) make the interpretation of the results less than straightforward. This casts the importance of the finding in doubt when it comes to increasing or understanding of the force-generating mechanism. This is contrary to what is stated by the authors who claim that the study “represents an important advance in our understanding of the nature of coupling between force generation and the release of the products of ATP-hydrolysis and thus how the chemical and mechanical events are coupled.” This statement from the discussion, repeated in even stronger terms in the abstract, is elaborated on to some extent in the final section of the Discussions before the conclusions. However, as stated above, the complexity of the data makes unambiguous interpretations difficult. Yet, the authors present a rather specific model for the relationship between Pi-release and

force-generation in the final section of the discussion (and Fig. 4C). This model is very similar to that

proposed previously by some of the co-authors (3, 9) with Pi-release preceding the power-stroke. More generally, this is probably the most popular model today (cf. 6, 10, 11). The idea of cross-bridge detachment into a post-power-stroke MADPPi state upon Pi-rebinding also seems to be part of the favored model in the present work. This idea is more unique to the present authors, but is only mentioned briefly and without explicit citation (as far as I could detect) to earlier studies of Debold and co-workers (e.g. 9) who first proposed this model.

A problem with models that assume Pi-release after the power-stroke is that they would predict reduced maximum steady-state velocity upon increased [Pi] (cf. 12), particularly if Pi-release is as slow as observed in experiments (4, 10, 11, 13). This is in contrast to experimental results. Moreover, such models would not account for a substantial monotonous reduction in steady-state isometric force (14, 12) found in experiments. Detachment into a post-power-stroke state upon Pi-rebinding would presumably take care of these problems with models assuming Pi-release after the power-stroke. However, whereas the proposed model with a detached post-power-stroke MADPPi state cannot be excluded based on the present data, there are difficulties with the idea. First, Sleep and Hutton (15) and later Bowater and Sleep (16) found that Pi-rebinding to myosin II leads to re-synthesis of ATP. Similar results were found (but with greatly differing kinetics) using myosin sub-fragment 1 and actin in solution (15) or isometrically contracting skinned muscle fibers (16). The mechanism proposed here, with Pi-induced detachment into a post-power-stroke state seems difficult to reconcile with these findings. Related to this argument, the overall evidence to support the existence of a post-power-stroke MADPPi state seems limited to results by only the present authors. Particularly, to the best of my knowledge, no structural evidence exists for a post-power-stroke MADPPi state. If evidence from other studies do exist, please cite them. I also wonder how the idea of Pi-induced detachment into a post-power-stroke state would be compatible with the data of Thomas and co-workers (10, 11) suggesting slow Pi-release in relation to the power-stroke? Further, is Pi assumed to bind to the active site also in the detached post-power-stroke state or to some other site?

Generally, I think it is premature to solely focus on models with Pi-release after the power-stroke and a specific model where Pi-induced detachment produces a post-power-stroke state with seemingly limited support in the literature and without full characterization of the model and its consequences. A credible model should account for most key phenomena from transient to steady-state mechanics of single molecules and ensembles as well as biochemical and structural data. If this cannot be achieved, as seems to be the case here, the least one can ask for is that the authors openly discuss different models rather than focusing on one favorite model.

2. Experimental complexities. First, the effects of the mutation S217A is, in itself, complex as elaborated on in greater detail below. Second, the use of mini-ensembles of myosin V instead of large ensembles or single molecules makes interpretation of the results challenging, e.g. prompting the discussion in the present paper on lines 237-248. More generally, in contrast to single molecules, it is not entirely clear if observed random events in the experiments are attributed to transitions within individual motors or to detachment/attachment events or even collective actions of motors (cf. 17). In my mind, large ensembles are also less ambiguous in this regard by more readily allowing assessment of average effects. These challenges may possibly be overcome by careful explanation of the data and thorough

analysis including use of Monte-Carlo simulation models. However, this has not been done here, leaving this reviewer somewhat puzzled about what the results actually mean.

The method of increasing load by increasing the trap stiffness, rather than applying a load-clamp, is somewhat indirect, adding to the complexity when it comes to easily getting a clear picture of what is going on.

With regard to the complex mutation effects, these have been carefully characterized in previous studies (1-3) showing effects of the S217A mutation on several biochemical transitions in the myosin cross-bridge cycle. For instance, the mutation increases the rate of ADP release, affects the re-priming of the lever arm in the recovery stroke following ATP binding and seems to affect binding of myosin to actin. The latter finding must be attributed to allosteric effects making it less obvious to infer a direct connection between S217A mutation and the strain-dependence of Pi-induced detachment. Moreover, in the kinetic analysis by Gunther et al. (2), no effect was found on the Pi-release step itself. In contrast Forgacs et al. (1), found an appreciable slowing of this step in myosin V as well as Llinas et al. (4) for the corresponding mutation in myosin VI. Whereas this may be explained by the higher ionic strength in the study of Gunther et al. (as pointed out in that study) it nevertheless points to uncertainties. Now, even if the mutation does prevent the entrance from the active site into the back door to slow Pi-release, it is not clear if the phosphate would stay in the active site for longer time due to this effect. Instead, the Pi may actually be trapped in an intermediate position similar to the secondary site proposed by Llinas et al. (4), elaborated on by Robert-Paganin et al. (18) and later incorporated into a complete kinetic model (12). Indeed, the corresponding serine in myosin VI (4) and myosin II (12) does not only communicate with the active site but also participates in Pi-coordination in the secondary Pi-binding site outside the active site. In relation to this notion, the effect of the S217A mutation on Pi-release did not prevent Houdusse, Sweeney and co-workers (observing this effect (4)) from sticking to their view that the Pi has to leave the active site before the power-stroke (4, 18, 19).

Finally, in relation to data on lines 157-167, one wonders if k_0 in the Bell equation reflects Pi-induced detachment only, without effects of ATP induced detachment. The latter possibility is a bit worrying for the S217A mutation as the binding life-time at low load is rather similar at 0 mM and 30 mM Pi (Fig. 2). Can it be excluded that this is due to a shift in detachment pathway from ATP induced detachment at low load to Pi-induced detachment at high loads? If both processes contribute in the mutant, as during relaxation after isometric contraction in myofibrils (8), this may at least partly contribute to the different load-dependence of the mutant compared to the wild-type myosin. In relation to this issue, I wonder about the Pi-concentration at 0 mM added Pi. Has this been measured? Additional control experiments may also be performed to elucidate possible conflation of Pi-induced and ATP-induced detachment.

3. Analysis. The quantitative analysis could be related to mechanistic models of muscle contraction rather than being largely phenomenological. First, the free energies of the different states are indicated vs an unspecified reaction coordinate. Second, the phenomenological quantitative treatment of the load dependence, using the Bell equation, verifies effects that are semi-quantitatively apparent without fitting of the equation to the data. It is thus not clear that this quantitative treatment, initially, developed for rather simple ligand-receptor interactions on cell surfaces (20), brings us closer to

understanding of the underlying molecular mechanisms in a complex case with different states that play central roles in active force generation. I am aware that the approach with the Bell equation has been used quite frequently in relation to studies of molecular motors, but I am not convinced about the real usefulness. I wonder if the formalism developed by Hill (21) and further by Eisenberg and Hill (22, 23) would be possible to use instead to provide more mechanistic insight.

Minor

Line 80 “..this class of molecular motors..” If talking of myosin in general, as seems to be the case here, I guess that "superfamily" should be used instead of "class".

Line 89. I wonder why a citation of Holmes and Geeves from 1999 is used instead of a more recent review paper, e.g. that of Houdusse, Sweeney and co-workers (18).

Line 106 “..a resistive load..” -> maybe “..an external resistive load..” for clarity.

Line 276. One may also consider some other studies in this connection (24, 8 25 7, 8, 26 27).

Lines 389-390. “Two-trap stiffness..” This sentence is not quite clear, I think.

Fig 2A and Fig. 3E-D.

i. The font of the text inside the figure is too small.

ii. It is not clear in the bar diagrams how many events were detected. Please state n and show all data superimposed on the bars or show box-and-whisker plots to indicate the distributions. If n is large and n as well as the variances are reasonably similar between groups ordinary parametric ANOVA could presumably be used instead with the benefit of increased power compared to non-parametric analysis.

Fig. 4. I presume that these experiments are performed at 30 mM Pi. However, this is not explicitly stated.

References

1. Forgacs, E. et al. Switch 1 mutation S217A converts myosin V into a low duty ratio motor. *J. Biol. Chem.* 284, 2138-2149 (2009).

2. Gunther, L.K. et al. FRET and optical trapping reveal mechanisms of actin activation of the power stroke and phosphate release in myosin V. *J. Biol. Chem.* 295, 17383-17397 (2020).
3. Scott, B., Marang, C., Woodward, M. & Debold, E.P. Myosin's powerstroke occurs prior to the release of phosphate from the active site. *Cytoskeleton (Hoboken)* (2021).
4. Llinas, P. et al. How actin initiates the motor activity of Myosin. *Dev Cell* 33, 401-412 (2015).
5. Sellers, J.R. & Veigel, C. Direct observation of the myosin-Va power stroke and its reversal. *Nature structural & molecular biology* 17, 590-595 (2010).
6. Woody, M.S., Winkelmann, D.A., Capitanio, M., Ostap, E.M. & Goldman, Y.E. Single molecule mechanics resolves the earliest events in force generation by cardiac myosin. *Elife* 8 (2019).
7. Wakefield, J.L, Bell, S.P. & Palmer, B.M. Inorganic phosphate accelerates cardiac myofilament relaxation in response to lengthening. *Front Physiol* 13, 980662 (2022).
8. Tesi, C., Piroddi, N., Colomo, F. & Poggesi, C. Relaxation kinetics following sudden Ca(2+) reduction in single myofibrils from skeletal muscle. *Biophys. J.* 83, 2142-2151 (2002).
9. Debold, E.P., Walcott, S., Woodward, M. & Turner, M.A. Direct observation of phosphate inhibiting the force-generating capacity of a miniensemble of Myosin molecules. *Biophys. J.* 105, 2374-2384 (2013).
10. Muretta, J.M., Rohde, J.A., Johnsrud, D.O., Cornea, S. & Thomas, D.D. Direct real-time detection of the structural and biochemical events in the myosin power stroke. *Proc. Natl. Acad. Sci. U. S. A.* 112, 14272-14277 (2015).
11. Trivedi, D.V. et al. Direct measurements of the coordination of lever arm swing and the catalytic cycle in myosin V. *Proc. Natl. Acad. Sci. U. S. A.* 112, 14593-14598 (2015).
12. Moretto, L. et al. Multistep orthophosphate release tunes actomyosin energy transduction. *Nature communications* 13, 4575 (2022).
13. White, H.D., Belknap, B. & Webb, M.R. Kinetics of nucleoside triphosphate cleavage and phosphate release steps by associated rabbit skeletal actomyosin, measured using a novel fluorescent probe for phosphate. *Biochemistry (Mosc).* 36, 11828-11836 (1997).
14. Smith, D.A. A new mechanokinetic model for muscle contraction, where force and movement are triggered by phosphate release. *J. Muscle Res. Cell Motil.* 35, 295-306 (2014).
15. Sleep, J.A. & Hutton, R.L. Exchange between inorganic phosphate and adenosine 5'-triphosphate in the medium by actomyosin subfragment 1. *Biochemistry (Mosc).* 19, 1276-1283 (1980).
16. Bowater, R. & Sleep, J. Demembranated muscle fibers catalyze a more rapid exchange between phosphate and adenosine triphosphate than actomyosin subfragment 1. *Biochemistry (Mosc).* 27, 5314-5323 (1988).
17. Hwang, Y., Washio, T., Hisada, T., Higuchi, H. & Kaya, M. A reverse stroke characterizes the force generation of cardiac myofilaments, leading to an understanding of heart function. *Proc. Natl. Acad. Sci. U. S. A.* 118 (2021).

18. Robert-Paganin, J., Pylypenko, O., Kikuti, C., Sweeney, H.L. & Houdusse, A. Force Generation by Myosin Motors: A Structural Perspective. *Chem. Rev.* 120, 5-35 (2020).
19. Planelles-Herrero, V.J., Hartman, J.J., Robert-Paganin, J., Malik, F.I. & Houdusse, A. Mechanistic and structural basis for activation of cardiac myosin force production by omecamtiv mecarbil. *Nature communications* 8, 190 (2017).
20. Bell, G.I. Models for the specific adhesion of cells to cells. *Science* 200, 618-627. (1978).
21. Hill, T.L. Theoretical formalism for the sliding filament model of contraction of striated muscle. Part I. *Prog. Biophys. Mol. Biol.* 28, 267-340 (1974).
22. Eisenberg, E. & Hill, T.L. A cross-bridge model of muscle contraction. *Prog. Biophys. Mol. Biol.* 33, 55-82 (1978).
23. Eisenberg, E., Hill, T.L. & Chen, Y. Cross-bridge model of muscle contraction. Quantitative analysis. *Biophys. J.* 29, 195-227 (1980).
24. Coupland, M.E., Puchert, E. & Ranatunga, K.W. Temperature dependence of active tension in mammalian (rabbit psoas) muscle fibres: effect of inorganic phosphate. *J Physiol* 536, 879-891 (2001).
25. Stehle, R., Kruger, M. & Pfitzer, G. Force kinetics and individual sarcomere dynamics in cardiac myofibrils after rapid Ca^{2+} changes. *Biophys. J.* 83, 2152-2161 (2002).
26. Stienen, G.J., Versteeg, P.G., Papp, Z. & Elzinga, G. Mechanical properties of skinned rabbit psoas and soleus muscle fibres during lengthening: effects of phosphate and Ca^{2+} . *J Physiol* 451, 503-523 (1992).
27. Bickham, D.C. et al. Millisecond-scale biochemical response to change in strain. *Biophys. J.* 101, 2445-2454 (2011).

Reviewer #2 (Remarks to the Author):

This is an exciting piece of work that addresses a long-standing issue in the myosin field the issue of Pi release from the actin.myosin motor and how it is coupled to force and movement. The paper is well written and the experiments well thought through. The work has several important conclusions – increased load leads to faster Pi rebinding consistent with earlier studies concluding that high loads slow the release of ADP from the attached crossbridge allowing Pi to rebind to the A.M.D crossbridge. The S217A mutant has been implicated in Pi release via the back door and here the mutant myosin is even more sensitive to load and the authors argue not easily compatible with Pi release before the power stroke. This is an important paper with novel observations using a new approach to studying Pi release and rebinding.

I have a few questions about the detailed measurements and the underlying assumptions. This may reflect my lack of familiarity with the details of myosin V mechanics, but these need to be explained to the average reader. These comments do not detract from this important and novel work

1. The work lacks the control (or a summary) of the effect of trap stiffness on events in the absence of added Pi. This is needed to fully understand the data presented and essential for the non-specialist
2. The key observation here is the number of heads interacting with actin since this is used in several calculations and this measurement needs a little unpicking. The author estimate the number of head available using STORM and the calculation based on the geometry comes out with a figure of “4-5 myosin available to interact with actin”. “Available to interact with actin” means what precisely. That at any one time the number of heads within range of the actin is 4-5 or that throughout the translocation of up to 40 nm this value is 4-5 in total. Do additional heads come into play or are the same 4-5 heads interacting with a different segment of actin (up to 40 nm translocated). This is now clear in the recently supplied Fig S2 but should be clearly stated in the main document
3. The assumption that a translocation of 25 – 40 nm means 4-5 heads each moving an average of 7 nm agrees with the 4-5 heads value above but assumes each head is making a single contribution of the same size to the movement. A missing figure is the number of events contributing to each translocation, are there 4-5 unitary events or is this an assumption. As events become shorter with added Pi are the events more frequent?
4. In Fig S2 the WT and S217A data appears as a single population of events (force and displacements) but as Pi and stiffness increases the distribution appears to change with a 2nd population of longer higher force events. Have the authors tried to analyse this as two populations?
5. The WT data is interpreted as higher force per crossbridge as the trap stiffness increased. In an ensemble assay in the absence of added Pi this would be interpreted as increase lifetime of attachments and therefore potentially more cross bridges contributing in the ensemble. Here the interpretation is that there is no change in the number of attached bridges – can this assumption be tested.

Minor

Fig 2 - the lettering superimposed on the darker bars of Fig 2A cannot be read.

Fig 4 legend is missing “Figure 4” at the start of the legend.

Fig 4C the dotted line implies the free energy well of the ADP.Pi post power stroke crossbridge is unaffected by resistive load – surely this is also higher. This may provide another explanation for faster Pi rebinding under load.

Fig S2. White on black labels on each fig are hard to read

P7 line 287 “Scott et al 2021” should be ref. 39

Reviewer #3 (Remarks to the Author):

The manuscript by Marang et al. describes an interesting set of single molecule experiments, giving new information on the force producing step in myosin motors and on the related biochemical states. The subject is timely and several papers have proposed opposing models in the last years. Therefore, I believe that the subject is of interest for the myosin research field and, more in general, for scientist working on molecular motors and for the readers of Nature Communications.

The manuscript is well written, the data are convincing and support the conclusions. However, there are in my opinion some major points that need to be addressed before the manuscript can be considered for publication in Nature Communications.

- 1) The authors increase the trap stiffness and observe a change in the kinetics of the actomyosin interaction. However, when the rigidity of the system increases, it is expected that the temporal resolution of the measurement increases as well so that shorter events can be detected. The authors should discuss this point and quantify to what extent the decrease in the average lifetime can be attributed to this effect.**
- 2) In the three bead assay, the average force applied on the filament is zero, independently of the trap stiffness. Depending on where the myosin molecule binds on the filament while the filament oscillates around zero force because of thermal forces and on the distance moved, the myosin molecule can experience both resistive and assistive forces. In a single interaction, several motors move the filament and some of them can experience assistive forces while others resistive forces. In the manuscript, this issue is oversimplified and only the peak force is considered, while I believe it should be discussed.**
- 3) Related to the previous point, in fig 4a the rate is plotted against peak force and fitted with Bell bond theory. However, the Bell bond theory assumes a constant force, which is not the case here. The authors should also consider plotting the detachment rate vs average force and/or the detachment rate of the last step vs peak force.**
- 4) The authors show how the lifetime changes with force in the presence of Pi, but not in the absence of Pi. In principle, this is important to understand if the observed changes in the lifetime are not only due to Pi-induced detachment, but also to the load dependence of myosin.ADP (or other states) detachment. Or are the experimental conditions such that Pi is always bound to myosin? What is the ATP concentration and the competition between ATP-induced detachment and Pi-induced detachment?**
- 5) In some sections of the manuscript (see for example lines 201-207, line 225-226) the authors assume that the load dependence that they observe is a consequence of load dependence of Pi rebinding. But it could also be that this is a consequence of the load dependence of the detachment rate of myosin.ADP.Pi, or a combination of the two.**

Minor points:

6) In the manuscript (lines 180-181), the authors assume that with a filament sliding of 29 nm and a step of the single head of 7 nm, 4 molecules are interacting. However, considering the high duty ratio of myosinVa, it is as well possible that 2 heads can displace the actin filament for long distances.

7) Lines 243-248 and Fig. 3F. It is possible that at low forces only two heads are actively moving the filament, while as the force increases the number of heads that interact with the filament increase.

8) Line 192: Fig. 4A shows the detachment rates plotted against the peak force

Response to Reviewers comments: Reviewers comments in italics, responses in normal font

Reviewer #1

This paper by Marang et al. presents results about the load dependence of Pi-release from the myosin active site and its relation to force generation. A three-bead optical tweezers system is used to probe the generation of force and displacements produced by wild-type myosin V motor domains and myosin V motor domains with an S217A mutation in switch I, a key nucleotide and Pi-binding loop in myosin. The studies are performed at varied concentration of inorganic phosphate, Pi, focusing on 30 mM Pi.

Specifically, the paper provides results on the load dependence of [Pi]-effects on force-generation by mini-ensembles of myosin V in the presence and absence of the mutation S217A. The authors find that external load increases the rate of actomyosin detachment due to Pi-binding. Moreover, based on data suggesting that the load-dependence of the detachment rate is increased in the presence of the S217A mutation the authors conclude that switch I is important for the force-dependent nature of Pi-induced detachment. The rationale of the study is clearly laid out in the Introduction. The Methods seem to be described in sufficient detail to repeat the experiments and the experiments appear to be carefully executed. The paper is also generally well written except for some minor issues raised below. However, I have some major concerns with the paper in other regards, particularly the extent to which the experimental data support the major conclusions:

1. This is one in a series of studies by the present authors and others of the effects of the S217A mutation in myosin V (1-3) and corresponding mutations in other myosins (4). The novelties presented here, compared to the previous studies, seem somewhat limited. This is for instance suggested by comparison to the recent paper by Scott and co-workers (3) who used optical tweezers to study motor function of myosin V in the presence of the S217A mutation and at varied Pi. What is new here compared to previous work would seem to be the increased sensitivity to external load of Pi-induced cross-bridge detachment in the presence of the S217A mutation. However, the load dependence of this detachment rate and its Pi dependence are known previously for wild type myosin (5, 6 7, 8).

- A. We appreciate the concern however we feel our work is highly novel because 1) only one of the papers cited above (5, Sellers and Veigel 2010) involved myosin Va. The others cited examined the effects of Pi on myosin II. And 2) the Sellers and Veigel paper was focused on identifying reversals of the powerstroke therefore the load-dependence of Pi rebinding was not explored to the same extent as in the present paper. Indeed, as stated in their paper the goal of the subset of experiments involving 10mM Pi was a control experiment to gain insight into the biochemical state of powerstroke reversals and recoveries, not to characterize the load-dependence of Pi rebinding. Their study was also a single molecule study where as ours was performed using mini-ensembles of myosin and thus yields novel information.

Whereas the increased load-sensitivity with the S217A mutation is an interesting finding, experimental complexities (summarized in the next point) make the interpretation of the results less than straightforward.

- B. The presence of this mutation changes the load-dependence of Pi-induced detachment from actin almost 7-fold. This finding is novel and independent of any model-dependent interpretation. Therefore, we feel this observation alone is quite novel and important to share with the field.

This casts the importance of the finding in doubt when it comes to increasing or understanding of the force-generating mechanism. This is contrary to what is stated by the authors who claim that the study "represents an important advance in our understanding of the nature of coupling between force generation and the release of the products of ATP-hydrolysis and thus how the chemical and mechanical events are coupled." This statement

from the discussion, repeated in even stronger terms in the abstract, is elaborated on to some extent in the final section of the Discussions before the conclusions. However, as stated above, the complexity of the data makes unambiguous interpretations difficult. Yet, the authors present a rather specific model for the relationship between Pi-release and force-generation in the final section of the discussion (and Fig. 4C). This model is very similar to that proposed previously by some of the co-authors (3, 9) with Pi-release preceding the power-stroke. More generally, this is probably the most popular model today (cf. 6, 10, 11). The idea of cross-bridge detachment into a post-power-stroke MADPPi state upon Pi-rebinding also seems to be part of the favored model in the present work. This idea is more unique to the present authors, but is only mentioned briefly and without explicit citation (as far as I could detect) to earlier studies of Debold and co-workers (e.g. 9) who first proposed this model.

- C. We have also edited the language regarding the impact of the findings for mechanism of coupling between force-generation product release to address this concern (see lines, 52,53 and 231).
- D. As a point of clarification, in our previous papers cited above (Scott et al. 2022 and Debold et al. 2013) we stated that our findings were most consistent with a model in which the powerstroke precedes Pi-release, not that Pi-release precedes the powerstroke as stated by the reviewer. This is also true of the authors of references 6, 10 and 11.
- E. Additionally, the fate of the cross-bridge following Pi-induced detachment is not a focus of the present report. Indeed, in the Discussion we only state that “A new P_i then rebound to actomyosin in a post-powerstroke, AM.ADP state, transiently creating an AM.ADP. P_i state that rapidly dissociated from actin to become an M.ADP. P_i state (Fig. 4C)”. This statement is consistent with both our findings and those of references 6,10 and 11.
 - a. We are however aware of the debate over whether reversals occur with Pi-rebinding and have addressed this issue in previous papers (Debold et al. 2011 and Debold et al. 2013). In the revised text we make it clear to the reader that this issue remains unclear, acknowledging both sides of the debate (see lines 363-364). However, we made Figure 4d consistent with our previous observations and models.

A problem with models that assume Pi-release after the power-stroke is that they would predict reduced maximum steady-state velocity upon increased [Pi] (cf. 12), particularly if Pi-release is as slow as observed in experiments (4, 10, 11, 13). This is in contrast to experimental results. Moreover, such models would not account for a substantial monotonous reduction in steady-state isometric force (14, 12) found in experiments. Detachment into a post-power-stroke state upon Pi-rebinding would presumably take care of these problems with models assuming Pi-release after the power-stroke.

- F. Although not the primary focus of the present paper this is indeed how we solved this issue in prior reports. We used a post-powerstroke detachment model to explain the effects of Pi on in vitro motility (Debold et al. 2011) and had expanded on the predictions of that model to account for the effects of Pi on force and steady state isometric ATPase data (Debold et al. 2013). References to these findings have been added to line 363-364 of the Discussion to highlight the debate over this issue.

However, whereas the proposed model with a detached post-power-stroke MADPPi state cannot be excluded based on the present data, there are difficulties with the idea. First, Sleep and Hutton (15) and later Bowater and Sleep (16) found that Pi-rebinding to myosin II leads to re-synthesis of ATP. Similar results were found (but with greatly differing kinetics) using myosin sub-fragment 1 and actin in solution (15) or isometrically contracting skinned muscle fibers (16). The mechanism proposed here, with Pi-induced detachment into a post-power-stroke state seems difficult to reconcile with these findings.

Related to this argument, the overall evidence to support the existence of a post-power-stroke MADPPi state seems limited to results by only the present authors. Particularly, to the best of my knowledge, no structural evidence exists for a post-power-stroke MADPPi state. If evidence from other studies do exist, please cite them. I also wonder how the idea of Pi-induced detachment into a post-power-stroke state would be compatible with the data of Thomas and co-workers (10, 11) suggesting slow Pi-release in relation to the power-stroke? Further, is Pi assumed to bind to the active site also in the detached post-power-stroke state or to some other site?

Generally, I think it is premature to solely focus on models with Pi-release after the power-stroke and a specific model where Pi-induced detachment produces a post-power-stroke state with seemingly limited support in the literature and without full characterization of the model and its consequences. A credible model should account for most key phenomena from transient to steady-state mechanics of single molecules and ensembles as well as biochemical and structural data. If this cannot be achieved, as seems to be the case here, the least one can ask for is that the authors openly discuss different models rather than focusing on one favorite model.

- G. Again, the issue of the fate of the cross-bridge following the detachment from a Pi-induced detachment is not a primary focus of manuscript. Rather we have other papers that have addressed this issue (e.g. Debold et al. 2013) in which we addressed the discrepancies with findings regarding the re-synthesis of ATP. And we have written a review paper discussing the current debate regarding the relative timing of Pi-release and the powerstroke (Debold, Cytoskeleton 2022). In the revised manuscript we acknowledge the that there is data, and papers, suggesting that Pi-release may occur prior to the powerstroke. We do so by expanding section of the Discussion detailing a scenario in which the present data might be consistent with this theory and we now only indicate that we favor an alternative model based on the present data (see lines 347-359).

2. Experimental complexities. First, the effects of the mutation S217A is, in itself, complex as elaborated on in greater detail below. Second, the use of mini-ensembles of myosin V instead of large ensembles or single molecules makes interpretation of the results challenging, e.g. prompting the discussion in the present paper on lines 237-248. More generally, in contrast to single molecules, it is not entirely clear if observed random events in the experiments are attributed to transitions within individual motors or to detachment/attachment events or even collective actions of motors (cf. 17). In my mind, large ensembles are also less ambiguous in this regard by more readily allowing assessment of average effects. These challenges may possibly be overcome by careful explanation of the data and thorough analysis including use of Monte-Carlo simulation models. However, this has not been done here, leaving this reviewer somewhat puzzled about what the results actually mean.

- A. We appreciate this concern and acknowledge the complexities of using small ensembles of myosin but we feel that our prior single molecule characterizations (Scott et al. 2021) and the new control data added (see Figs 1-4) enable a detailed, and useful, interpretation of these data. Specifically, the fact that we had previously quantified the single molecule step size of both constructs (Scott et al. Cytoskeleton 2021) enables us to estimate the number of single molecule displacements during each binding event (see Fig. 3d and 3f). Second, the additional control data obtained in the absence of added Pi (at the suggestion of Reviewers 2 and 3), show that increasing resistive load in the absence of Pi prolong the attachment time. This confirms that the increased detachment rate in the presence of Pi is due to a Pi-induced detachment and mostly likely occurs from an AM.ADP.Pi bound state. Thus, we feel that these data can be interpreted and provide useful insight into the mechanism of load-dependent Pi-induced detachment.

Regarding the modelling of these data, the construction and testing of a detailed molecular model is a goal of ours, and in this effort we prefer to take a highly detailed approach as we have done in the past (see e.g. Walcott et al. Biophysical Journal 2012). This will require a significant effort, especially if, as suggested, we compare and contrast competing models on the relative timing of Pi-release and the powerstroke. Indeed, it is typical for such efforts to encompass a full manuscript (see e.g. Mansson 2019 and Ranatunga and Offer 2020) therefore such an effort is beyond the scope of the current experimental paper.

The method of increasing load by increasing the trap stiffness, rather than applying a load-clamp, is somewhat indirect, adding to the complexity when it comes to easily getting a clear picture of what is going on.

- B. We understand the concern however we actually believe this approach has distinct advantages over a load-clamp assay. Firstly, because in the typical load-clamped laser trap assay (e.g. Veigel and Molloy NSMB 2003) there is a significant delay before the load is applied. This is because the binding event must first be detected before the load is applied and then there are software and hardware delays to overcome before the desired load is reached. These can result in delays of 10ms or more before the load is applied and thus the load is not felt by myosin until after the completion of the powerstroke and Pi release, and in the present experiment this would have likely also occurred after some amount of Pi-rebinding and detachment. By contrast by simply increasing the stiffness the myosin experiences the increased load immediately upon binding to the actin filament. This is an advantage that was ideal for the present set of experiments where we wanted to see the effects of the powerstroke and Pi-induced detachment from actin. And we believe we can provide a coherent interpretation of the data as detailed above (Response to point 2a).

With regard to the complex mutation effects, these have been carefully characterized in previous studies (1-3) showing effects of the S217A mutation on several biochemical transitions in the myosin cross-bridge cycle. For instance, the mutation increases the rate of ADP release, affects the re-priming of the lever arm in the recovery stroke following ATP binding and seems to affect binding of myosin to actin. The latter finding must be attributed to allosteric effects making it less obvious to infer a direct connection between S217A mutation and the strain-dependence of Pi-induced detachment. Moreover, in the kinetic analysis by Gunther et al. (2), no effect was found on the Pi-release step itself. In contrast Forgacs et al. (1), found an appreciable slowing of this step in myosin V as well as Llinas et al. (4) for the corresponding mutation in myosin VI. Whereas this may be explained by the higher ionic strength in the study of Gunther et al. (as pointed out in that study) it nevertheless points to uncertainties.

- C. We are aware of the prior investigations using this mutation and including our report including the accelerated rate of ADP-release in solution (Gunther et al. 2020). Indeed, we expected because of the increase in this rate that it would be less susceptible to Pi-induced detachment because it would get through the state vulnerable to Pi-rebinding more quickly than WT. Indeed, the observations at low trap stiffness were consistent with this notion. However, to our surprise as the load increased S217A became more susceptible to Pi-induced detachment than WT. This does not seem to be a confounder but rather highlights the greater load-sensitivity.

Regarding the effects on the recovery stroke, such an effect would not impact the present findings as the states off actin, where the recovery stroke occurs, are not visible or perturbed in the laser trap assay. Similarly, weak-binding is not detected in our laser trap assay thus

slowed weak-to-strong transition observed for this mutant would not affect the present results as we are only characterizing the effect of load after this transition is completed.

In Gunther et al. 2020 we could not definitively determine whether Pi-release was slowed by the mutation because at the ionic strength used we could not reach an actin concentration that could saturate this rate. See the data below re-plotted from Gunther et al. 2020 here:

"[Redacted]"

Thus, the more definitive experiments for determining the effect of on Pi-release were performed by Forgacs et al. 2009 because they used an ionic strength low enough to obtain a saturating concentration of actin. They reported a 10-fold decrease in the rate of Pi release in this mutation at the saturating actin concentration, which strongly suggests that the presence of this mutation does indeed slow the rate of actin-dependent Pi-release. The same conclusion was reached by Llinas et al. 2015 who characterized the same mutation in myosin Va, as well as the corresponding mutation in myosin II and VI. And they indicated that in myosin II the result was obtained at low ionic strength, suggesting that it was slowed at a saturating actin concentration.

Now, even if the mutation does prevent the entrance from the active site into the back door to slow Pi-release, it is not clear if the phosphate would stay in the active site for longer time due to this effect. Instead, the Pi may actually be trapped in an intermediate position similar to the secondary site proposed by Llinas et al. (4), elaborated on by Robert-Paganin et al. (18) and later incorporated into a complete kinetic model (12). Indeed, the corresponding serine in myosin VI (4) and myosin II (12) does not only communicate with the active site but also participates in Pi-coordination in the secondary Pi-binding site outside the active site. In relation to this notion, the effect of the S217A mutation on Pi-release did not prevent Houdusse, Sweeney and co-workers (observing this effect (4)) from sticking to their view that the Pi has to leave the active site before the power-stroke (4, 18, d19).

- D. In Llinas et al. 2015 they state that the S217A mutation “was designed to slow Pi entry of the Pi into the tunnel,” and in Table 1 of the same paper they list this mutation as “Impeding Pi entry into the tunnel seen in the PiR structure.” This is in contrast to the mutations they introduced to impede exit of Pi from the tunnel (e.g. Y439E in myosin Va). The view that this mutation would alter release from the active site is also consistent with the structure and conclusions reported by Smith and Rayment (1996), and later others (Reubold et al. 2003

and Coureux et al. 2004), indicating that this conserved residue “is within hydrogen-bonding distance of the oxygen of vanadate moiety of $MgADP \cdot VO_4$ ”, an analog of the M.ADP.Pi state. Thus, it would seem to us that the current consensus is that this mutation impedes the entry of Pi out of the active site and into the tunnel, rather than causing it to stall outside of the active site in some secondary binding site.

However, this perceived ambiguity is exactly why we also examined the effects of elevated phosphate as an independent approach to maintaining Pi in the active site. Under the conditions in the present study (30mM added Pi and increasing resistive load), Pi rapidly rebinds to the active site. The pronounced reduction in the binding event lifetimes (Fig. 2) provides strong evidence that Pi does indeed rapidly rebind to the active site, and induce detachment from actin, as this likely induced opening of the actin-binding cleft and decreased the affinity for actin.

A final point to make is that the focus of the current paper is on the load-dependence of rebinding and not the order of release. The effects on the rate of rebinding are not dependent on a specific order of release rather it is on the mechanisms of Pi-induced dissociation and are therefore not dependent on the specific effect of the mutation on the nature of Pi release from the active site.

Finally, in relation to data on lines 157-167, one wonders if k_0 in the Bell equation reflects Pi-induced detachment only, without effects of ATP induced detachment. The latter possibility is a bit worrying for the S217A mutation as the binding life-time at low load is rather similar at 0 mM and 30 mM Pi (Fig. 2). Can it be excluded that this is due to a shift in detachment pathway from ATP induced detachment at low low to Pi-induced detachment at high loads? If both processes contribute in the mutant, as during relaxation after isometric contraction in myofibrils (8), this may at least partly contribute to the different load-dependence of the mutant compared to the wild-type myosin. In relation to this issue, I wonder about the Pi-concentration at 0 mM added Pi. Has this been measured? Additional control experiments may also be performed to elucidate possible conflation of Pi-induced and ATP-induced detachment.

- E. We agree with this observation, k_0 theoretically reflects the rate of Pi-induced detachment in the absence of Pi, but detachment would still occur via ATP-induced dissociation as stated. Indeed, our stated interpretation of this effect is that Pi does not readily rebind to the mutant construct in the absence of load, therefore the detachment rate (k_0) would reflect the ATP-induced detachment rate. We have now indicated this to the reader in the revised manuscript (see lines 304-305).

The Pi concentration was not directly measured in the 0mM added condition, this is why we refer to it as “0mM added P”_i. Estimates have placed the level of contamination in a 0 added Pi have ranged from a 100 to 800uM (Millar and Homsher 1990, Dantzig et al. 1992), however these estimates are from single muscle fiber experiments that include orders of magnitude more myosin and actin. More importantly their buffers contained creatine phosphate and creatine kinase, which are not in our buffers. Indeed, the largest source of contaminating Pi is due to the creatine kinase reaction in muscle fibers (Dantzig et al. 1992). Therefore, the amount of contamination would be much lower in the present study, likely less than 100uM. This low estimated value would be consistent with the new additional control experiments in the absence of added Pi showing that the binding event lifetimes increase with more resistive load rather decrease as in the presence of Pi (see Fig. 2), strongly suggesting that very little Pi is around to rebind at 0 added Pi. Thus, Pi contamination likely had only a minuscule, if any, effect on the present findings.

3. Analysis. The quantitative analysis could be related to mechanistic models of muscle contraction rather than being largely phenomenological. First, the free energies of the different states are indicated vs an unspecified reaction coordinate. Second, the phenomenological quantitative treatment of the load dependence, using the Bell equation, verifies effects that are semi-quantitatively apparent without fitting of the equation to the data. It is thus not clear that this quantitative treatment, initially, developed for rather simple ligand-receptor interactions on cell surfaces (20), brings us closer to understanding of the underlying molecular mechanisms in a complex case with different states that play central roles in active force generation. I am aware that the approach with the Bell equation has been used quite frequently in relation to studies of molecular motors, but I am not convinced about the real usefulness. I wonder if the formalism developed by Hill (21) and further by Eisenberg and Hill (22, 23) would be possible to use instead to provide more mechanistic insight.

3A. We are hesitant to use models developed on muscle to interpret these data from myosin Va. The kinetics have been shown to be quite distinct from muscle myosin and therefore we would be more comfortable constructing models using myosin Va specific measures and models (see e.g. Trybus et al. 1999 and Del la Cruz et al. 1999). Indeed, as stated above, it is our intention to do so in a future paper dedicated to a broader mechanistic effort, but such an endeavor is beyond the scope of the current paper.

B. Based on a similar concern regarding the use of the Bell equation and its interpretation in the present paper by Reviewer 3, we have refined the approach such that we examine the lifetime of only the last step and thus at a constant force (see Figure 4a). We are aware of the debate and concerns over the interpretation of the parameters of the Bell equation in a more complex system of molecular motors (See e.g. Walcott 2008). However, even if the data cannot be strictly interpreted in terms of transition-state theory (i.e. the distance to the transition state) the *d*-value serves as a parameter that is a quantitative indicator of degree of load-dependence independent of transition state theory. Figure 4d is meant only as an illustrative cartoon of our qualitative interpretation of how load affects the rate of Pi-induced detachment, which is similar to many other papers using the Bell-equation (e.g. Veigel and Molloy 2003, Sellers and Veigel 2010). We depict load as tilting the energy landscape, which is a widely accepted and useful interpretation of how load affects a rate constant (see Chapter 5 of Jo Howard's text *Mechanics of Motor Proteins and the Cytoskeleton*), therefore we feel it is a useful and important idea to share with the reader.

Minor

Line 80 *"..this class of molecular motors.." If talking of myosin in general, as seems to be the case here, I guess that "superfamily" should be used instead of "class".*

Changed as suggested (see line 81)

Line 89. *I wonder why a citation of Holmes and Geeves from 1999 is used instead of a more recent review paper, e.g. that of Houdusse, Sweeney and co-workers (18).*
Line 106 *"..a resistive load.." -> maybe *"..an external resistive load.."* for clarity.*

Reference added as suggested, and edit made as suggested (see lines 90 and 107)

Line 276. One may also consider some other studies in this connection (24, 8 25 7, 8, 26 27).

The statement made refers to the opposing effects on maximal unloaded shortening velocity and maximal isometric force, so we referenced this study because both were measured. In the suggested references typically only force is emphasized or cardiac preparations were used, so it doesn't seem to appropriately support the present statement.

Lines 389-390. "Two-trap stiffness.." This sentence is not quite clear, I think.

Edited to make it clearer (see line 425)

Fig 2A and Fig. 3E-D.

i. The font of the text inside the figure is too small.

ii. It is not clear in the bar diagrams how many events were detected. Please state n and show all data superimposed on the bars or show box-and-whisker plots to indicate the distributions. If n is large and n as well as the variances are reasonably similar between groups ordinary parametric ANOVA could presumably be used instead with the benefit of increased power compared to non-parametric analysis.

Font was enlarged as suggested.

Sample size for each condition has been added to the Figure 2 caption.

The data failed a test of Normality by a large margin ($p < 0.0001$) and the sample sizes are quite different among the conditions (see caption for n) therefore we felt it was most appropriate to stick with the non-parametric analyses.

Fig. 4. I presume that these experiments are performed at 30 mM Pi. However, this is not explicitly stated.

Indicated in the revised caption as suggested.

References

1. Forgacs, E. et al. Switch 1 mutation S217A converts myosin V into a low duty ratio motor. *J. Biol. Chem.* 284, 2138-2149 (2009).
2. Gunther, L.K. et al. FRET and optical trapping reveal mechanisms of actin activation of the power stroke and phosphate release in myosin V. *J. Biol. Chem.* 295, 17383-17397 (2020).
3. Scott, B., Marang, C., Woodward, M. & Debold, E.P. Myosin's powerstroke occurs prior to the release of phosphate from the active site. *Cytoskeleton (Hoboken)* (2021).
4. Llinas, P. et al. How actin initiates the motor activity of Myosin. *Dev Cell* 33, 401-412 (2015).
5. Sellers, J.R. & Veigel, C. Direct observation of the myosin-Va power stroke and its reversal. *Nature structural & molecular biology* 17, 590-595 (2010).
6. Woody, M.S., Winkelmann, D.A., Capitanio, M., Ostap, E.M. & Goldman, Y.E. Single molecule mechanics resolves the earliest events in force generation by cardiac myosin. *Elife* 8 (2019).

7. Wakefield, J.I., Bell, S.P. & Palmer, B.M. Inorganic phosphate accelerates cardiac myofilament relaxation in response to lengthening. *Front Physiol* 13, 980662 (2022).
8. Tesi, C., Piroddi, N., Colomo, F. & Poggesi, C. Relaxation kinetics following sudden Ca(2+) reduction in single myofibrils from skeletal muscle. *Biophys. J.* 83, 2142-2151 (2002).
9. Debold, E.P., Walcott, S., Woodward, M. & Turner, M.A. Direct observation of phosphate inhibiting the force-generating capacity of a miniensemble of Myosin molecules. *Biophys. J.* 105, 2374-2384 (2013).
10. Muretta, J.M., Rohde, J.A., Johnsrud, D.O., Cornea, S. & Thomas, D.D. Direct real-time detection of the structural and biochemical events in the myosin power stroke. *Proc. Natl. Acad. Sci. U. S. A.* 112, 14272-14277 (2015).
11. Trivedi, D.V. et al. Direct measurements of the coordination of lever arm swing and the catalytic cycle in myosin V. *Proc. Natl. Acad. Sci. U. S. A.* 112, 14593-14598 (2015).
12. Moretto, L. et al. Multistep orthophosphate release tunes actomyosin energy transduction. *Nature communications* 13, 4575 (2022).
13. White, H.D., Belknap, B. & Webb, M.R. Kinetics of nucleoside triphosphate cleavage and phosphate release steps by associated rabbit skeletal actomyosin, measured using a novel fluorescent probe for phosphate. *Biochemistry (Mosc).* 36, 11828-11836 (1997).
14. Smith, D.A. A new mechanokinetic model for muscle contraction, where force and movement are triggered by phosphate release. *J. Muscle Res. Cell Motil.* 35, 295-306 (2014).
15. Sleep, J.A. & Hutton, R.L. Exchange between inorganic phosphate and adenosine 5'-triphosphate in the medium by actomyosin subfragment 1. *Biochemistry (Mosc).* 19, 1276-1283 (1980).
16. Bowater, R. & Sleep, J. Demembrated muscle fibers catalyze a more rapid exchange between phosphate and adenosine triphosphate than actomyosin subfragment 1. *Biochemistry (Mosc).* 27, 5314-5323 (1988).
17. Hwang, Y., Washio, T., Hisada, T., Higuchi, H. & Kaya, M. A reverse stroke characterizes the force generation of cardiac myofilaments, leading to an understanding of heart function. *Proc. Natl. Acad. Sci. U. S. A.* 118 (2021).
18. Robert-Paganin, J., Pylypenko, O., Kikuti, C., Sweeney, H.L. & Houdusse, A. Force Generation by Myosin Motors: A Structural Perspective. *Chem. Rev.* 120, 5-35 (2020).
19. Planelles-Herrero, V.J., Hartman, J.J., Robert-Paganin, J., Malik, F.I. & Houdusse, A. Mechanistic and structural basis for activation of cardiac myosin force production by omecamtiv mecarbil. *Nature communications* 8, 190 (2017).
20. Bell, G.I. Models for the specific adhesion of cells to cells. *Science* 200, 618-627. (1978).
21. Hill, T.L. Theoretical formalism for the sliding filament model of contraction of striated muscle. Part I. *Prog. Biophys. Mol. Biol.* 28, 267-340 (1974).
22. Eisenberg, E. & Hill, T.L. A cross-bridge model of muscle contraction. *Prog. Biophys. Mol. Biol.* 33, 55-82 (1978).
23. Eisenberg, E., Hill, T.L. & Chen, Y. Cross-bridge model of muscle contraction. Quantitative analysis. *Biophys. J.* 29, 195-227 (1980).
24. Coupland, M.E., Puchert, E. & Ranatunga, K.W. Temperature dependence of active tension in mammalian (rabbit psoas) muscle fibres: effect of inorganic phosphate. *J Physiol* 536, 879-891 (2001).
25. Stehle, R., Kruger, M. & Pfitzer, G. Force kinetics and individual sarcomere dynamics in cardiac myofibrils after rapid ca(2+) changes. *Biophys. J.* 83, 2152-2161 (2002).
26. Stienen, G.J., Versteeg, P.G., Papp, Z. & Elzinga, G. Mechanical properties of skinned rabbit psoas and soleus muscle fibres during lengthening: effects of

phosphate and Ca²⁺. J Physiol 451, 503-523 (1992).

27. Bickham, D.C. et al. Millisecond-scale biochemical response to change in strain. Biophys. J. 101, 2445-2454 (2011).

Reviewer #2 (Remarks to the Author):

This is an exciting piece of work that addresses a long-standing issue in the myosin field the issue of Pi release from the actin.myosin motor and how it is coupled to force and movement. The paper is well written and the experiments well thought through. The work has several important conclusions – increased load leads to faster Pi rebinding consistent with earlier studies concluding that high loads slow the release of ADP from the attached crossbridge allowing Pi to rebind to the A.M.D crossbridge. The S217A mutant has been implicated in Pi release via the back door and here the mutant myosin is even more sensitive to load and the authors argue not easily compatible with Pi release before the power stroke. This is an important paper with novel observations using a new approach to studying Pi release and rebinding.

I have a few questions about the detailed measurements and the underlying assumptions. This may reflect my lack of familiarity with the details of myosin V mechanics, but these need to be explained to the average reader. These comments do not detract from this important and novel work

1. The work lacks the control (or a summary) of the effect of trap stiffness on events in the absence of added Pi. This is needed to fully understand the data presented and essential for the non-specialist

This was an excellent suggestion. Therefore we added control datasets in the absence of added Pi for both the wild-type and mutant constructs as suggested (See Figs 1-4 and lines 145-147). This has added an important piece to the paper that confirms and accentuates the load-dependent effect of Pi. In our view this greatly improves the impact of the findings, thank you so much for this suggestion.

2. The key observation here is the number of heads interacting with actin since this is used in several calculations and this measurement needs a little unpicking. The author estimate the number of head available using STORM and the calculation based on the geometry comes out with a figure of "4-5 myosin available to interact with actin". "Available to interact with actin" means what precisely. That at any one time the number of heads withing range of the actin is 4-5 or that throughout the translocation of up to 40 nm this value is 4-5 in total. Do additional heads come into play or are the same 4-5 heads interacting with a different segment of actin (up to 40 nm translocated). This is now clear in the recently supplied Fig S2 but should be clearly stated in the main document

Sorry for the confusion. Yes, this reflects the number of myosin molecules within reach of the actin filament. A clarification has been added to the main text in the revised version (see lines 466-470).

3. The assumption that a translocation of 25 – 40 nm means 4-5 heads each moving an average of 7 nm agrees with the 4-5 heads value above but assumes each head is making a single contribution of the same size to the movement. A missing figure is the number of events contributing to each translocation, are there 4-5 unitary

events or is this an assumption. As events become shorter with added Pi are the events more frequent?

This is an assumption, but relies on our previous direct measurement of the single molecule step size of both constructs (Scott et al. 2021). We now indicate that this is an assumption and assumes that each head contributes the same size step (see lines 185-186). Unfortunately, we cannot detect individual events within a run because at the ATP concentration used (100uM) the events are faster than the temporal resolution of our detection method so we cannot determine the frequency of myosin attachments within a binding event.

4. In Fig S2 the WT and S217A data appears as a single population of events (force and displacements) but as Pi and stiffness increases the distribution appears to change with a 2nd population of longer higher force events. Have the authors tried to analyse this as two populations?

This was an excellent thought however, upon further examination much of the difference in shape of the distributions was due to the bin width chosen for the distributions, and a uniform rightward shift due to the increased stiffness. To address this issue we now present these data as a cumulative distribution (see S2), which is not dependent on bin width. There does seem to be an effect but it is not clear to use how to interpret that effect. We feel that the effect on the event lifetime presents a clearer explanation of how Pi is affecting the myosin so we have not interpreted these observations any further at present. This may however be a useful exercise in a future modelling effort.

5. The WT data is interpreted as higher force per crossbridge as the trap stiffness increased. In an ensemble assay in the absence of added Pi this would be interpreted as increase lifetime of attachments and therefore potentially more cross bridges contributing in the ensemble. Here the interpretation is that there is no change in the number of attached bridges – can this assumption be tested.

This interpretation was based on the assumption that each myosin head generates a 7nm step upon binding to the actin filament. Since the stiffness was increased by more than the displacements, as Pi was added, this resulted in an increase in the predicted force per cross-bridge. There is no way to directly test this assumption based on the present data. Theoretically, the change in the noise upon binding of additional heads to actin could be used, but the change in the signal is insignificant from 1 to 2, 3 or more heads bound.

Minor

Fig 2 - the lettering superimposed on the darker bars of Fig 2A cannot be read.

Corrected in the revised Figure 2

Fig 4 legend is missing "Figure 4" at the start of the legend.

Corrected in revised Figure 4

Fig 4C the dotted line implies the free energy well of the ADP.Pi post power stroke crossbridge is unaffected by resistive load – surely this is also higher. This may provide another explanation for faster Pi rebinding under load.

We agree and in the revised Figure (4d) we tilted the entire landscape to reflect this notion.

Fig S2. White on black labels on each fig are hard to read

Font enlarged in the revised text (See Fig. S1)

P7 line 287 "Scott et al 2021" should be ref. 39

Corrected in revised text, see line 293

Reviewer #3 (Remarks to the Author):

The manuscript by Marang et al. describes an interesting set of single molecule experiments, giving new information on the force producing step in myosin motors and on the related biochemical states. The subject is timely and several papers have proposed opposing models in the last years. Therefore, I believe that the subject is of interest for the myosin research field and, more in general, for scientist working on molecular motors and for the readers of Nature Communications. The manuscript is well written, the data are convincing and support the conclusions. However, there are in my opinion some major points that need to be addressed before the manuscript can be considered for publication in Nature Communications.

1) The authors increase the trap stiffness and observe a change in the kinetics of the actomyosin interaction. However, when the rigidity of the system increases, it is expected that the temporal resolution of the measurement increases as well so that shorter events can be detected. The authors should discuss this point and quantify to what extent the decrease in the average lifetime can be attributed to this effect.

This is correct and our potential temporal resolution was likely improved at the higher stiffness (~2ms to ~1.5ms, based on the roll off value from an FFT on the bead motion), however our actual temporal resolution is limited to a much greater extent by the method of detection of the binding events (HMM and Changeoint analysis, see Methods), which is only able to detect events that are ~5-10ms in duration. More importantly the new control data show that increasing the trap stiffness in the absence of Pi actually increases the event lifetimes (Fig. 2), demonstrating that the increased stiffness alone did not cause the dramatic decreases in event lifetimes in the presence of Pi.

2) In the three bead assay, the average force applied on the filament is zero, independently of the trap stiffness. Depending on where the myosin molecule binds on the filament while the filament oscillates around zero force because of thermal forces and on the distance moved, the myosin molecule can experience both resistive and assistive forces. In a single interaction, several motors move the filament and some of them can experience assistive forces while others resistive forces. In the manuscript, this issue is oversimplified and only the peak force is considered, while I believe it should be discussed.

Yes, this is correct. We only examined runs in the positive direction. And we have added a statement to acknowledge the complexity of mini-ensembles of myosin as suggested (see lines 184-188). However, this issue is greatly minimized now that we have taken your suggestion and re-analyzed the data to only look at the final step in the traces, (see, below and Figure 4a). Thank you again for that suggestion.

3) Related to the previous point, in fig 4a the rate is plotted against peak force and fitted with Bell bond theory. However, the Bell bond theory assumes a constant force, which is not the case here. The authors should also consider plotting the detachment rate vs average force and/or the detachment rate of the last step vs peak force.

Thank you for this suggestion, we re-analyzed the data in exactly this way and now this Figure (4a) reflects the resultant values to enable a more appropriate interpretation of the parameters of the Bell equation.

4) The authors show how the lifetime changes with force in the presence of Pi, but not in the absence of Pi. In principle, this is important to understand if the observed changes in the lifetime are not only due to Pi-induced detachment, but also to the load dependence of myosin.ADP (or other states) detachment. Or are the experimental conditions such that Pi is always bound to myosin? What is the ATP concentration and the competition between ATP-induced detachment and Pi-induced detachment?

We added the control datasets for both WT and the mutant, which confirm that the effect on the detachment rate is due to Pi (see Figs 1-4). This suggestion makes the original findings much more impactful, thank again for suggesting this addition.

5) In some sections of the manuscript (see for example lines 201-207, line 225-226) the authors assume that the load dependence that they observe is a consequence of load dependence of Pi rebinding. But it could also be that this is a consequence of the load dependence of the detachment rate of myosin.ADP.Pi, or a combination of the two.

The control data we added at your suggestion demonstrates that the increased stiffness and resultant increase in resistive load slows the detachment rate in the absence of Pi. Therefore, this strongly suggests that the accelerated detachment rate in the presence of Pi and high trap stiffness was due to the load-dependence of Pi-rebinding. Again, thank you for the suggestion of adding this control, it makes this result much clearer.

Minor points:

6) In the manuscript (lines 180-181), the authors assume that with a filament sliding of 29 nm and a step of the single head of 7 nm, 4 molecules are interacting. However, considering the high duty ratio of myosinVa, it is as well possible that 2 heads can displace the actin filament for long distances.

Yes, that is true therefore we make it clear in the revised text that this is based on the assumptions of our analysis (see lines 182-189).

7) Lines 243-248 and Fig. 3F. It is possible that at low forces only two heads are actively moving the filament, while as the force increases the number of heads that interact with the filament increase.

Yes, so we now make it clearer to the reader our assumption that each head only interacts once with the myosin during an event (see lines 252 and 253). A future modelling paper will allow us to explore these potential mechanisms more fully.

8) Line 192: Fig. 4A shows the detachment rates plotted against the peak force

Corrected as suggested (see lines 202-203)

REVIEWERS' COMMENTS

Reviewer #1 (Remarks to the Author):

I can mostly accept the argumentation by the authors and I think that the changes made to the manuscript have led to significant improvements.

However, I still have some critical problems with the manuscript that need to be further considered.

1. A major issue is that, with Pi-release after the power-stroke that is as slow as found recently (e.g. from the Thomas group), there must be a mechanism to account for the high and [Pi] insensitive sliding velocity. Without the escape route into a post-power-stroke state upon Pi-rebinding I cannot see how the authors can come up with such a mechanism. Therefore it is not quite appropriate to just state that the post-power-stroke detachment is not in primary focus here and then leave the problem. In contrast, this issue (or an alternative rescuing mechanism) is strongly intertwined with the general credibility of models with Pi-release after the power-stroke. In this context, I also noted that the authors did not explicitly respond to/comment on my remarks that models with Pi-release after the power-stroke predict reduced maximum velocity.

Now, one may of course claim that this study is about myosin V whereas the effects on velocity have primarily been seen with myosin II. However, there is evidence from different studies (e.g. Muretta et al vs Trivedi et al PNAS 2015a,b and Moretto et al 2022 vs Llinas et al 2015) that the Pi-release mechanism is similar in different myosin. Indeed, also Debold and co-workers initially developed their model for myosin II whereas then applying it (or an easily recognizable version of it) to myosin V. Finally, in the present manuscript they add to this notion by claiming "load sensitivity of Pi-rebinding to the nucleotide-binding site may be conserved across several members of the myosin family."

In view of the above, I think that the least one can require is that the authors discuss how the idea of Pi-release after the power-stroke can be made consistent with a high maximum gliding velocity (of myosin II or V) that is also (largely) [Pi] insensitive (at least without velocity decreasing with increased [Pi]).

2. In the section "Implications for transduction"

Lines 333-335: "Therefore, the rebinding of Pi to the active site should prevent the powerstroke and thus prevent or inhibit displacements as load is increased, because Pi will more readily rebind to the active site at higher resistive loads."

This may be incorrectly phrased. An alternative that, as far as I see, cannot be excluded is that Pi may bind with equal probability (rate constant) at different loads but only leads to detachment at high load.

A mechanism of that type was proposed by Moretto et al (2022) but it seems to me, unless I have misunderstood something, that the authors have misinterpreted that model. Thus, in the model of Moretto et al., due to equilibrium between Pi at the second site and the active site (Fig. 4a of Moretto et al), allosteric effects of Pi-rebinding on actomyosin affinity is most likely mediated by occupation of the active site also in this case. The increased Pi-induced detachment rate at increased load per myosin head would follow in the multistep model of Moretto et al. from the free energy diagrams (extracted from their paper in figure 1, attached). In that model, the increased probability of detachment with increased load would be a lower difference in free energy between the detached MADPPi state and the AMADP state (or AMADPP' state) at higher load per myosin head (thick arrow; higher x-value in figure above). This follows from the location of the different free energy diagrams (for pre and post-powerstroke states) along the x-axis and the axis for free energy.

Minor

Lines 318-319 Statement “This seems the more likely explanation given that the substitution is in switch I.”

It may be appropriate to mention that Llinas et al, (2015) who also observed effects of this mutation (together with several others) on Pi-release came up with another interpretation.”

Reviewer #2 (Remarks to the Author):

This is a revised manuscript in which the authors have address each of the issues raised in my original review.

Reviewer #3 (Remarks to the Author):

The authors addressed all my comments. The new data and analysis greatly enhanced the manuscript and I suggest to publish it in Nature Communications. There is just a minor issue that I suggest to fix before publication:

- In the revised manuscript the detachment rate plotted in Fig.4b is obtained from the time spent at peak force Tpf, as shown in Fig.4a. It is not clear from the figure and the caption how Tpf is determined, since the beginning of Tpf might be somehow arbitrary. Have the authors used some kind of threshold to set the beginning of Tpf? I suggest to add a paragraph in the methods to explain this point.

Response to second round of reviews

Reviewers comments in italics, responses in plain text.

Reviewer 1:

I can mostly accept the argumentation by the authors and I think that the changes made to the manuscript have led to significant improvements.

However, I still have some critical problems with the manuscript that need to be further considered.

1. A major issue is that, with Pi-release after the power-stroke that is as slow as found recently (e.g. from the Thomas group), there must be a mechanism to account for the high and [Pi] insensitive sliding velocity. Without the escape route into a post-power-stroke state upon Pi-rebinding I cannot see how the authors can come up with such a mechanism. Therefore it is not quite appropriate to just state that the post-power-stroke detachment is not in primary focus here and then leave the problem. In contrast, this issue (or an alternative rescuing mechanism) is strongly intertwined with the general credibility of models with Pi-release after the power-stroke. In this context, I also noted that the authors did not explicitly respond to/comment on my remarks that models with Pi-release after the power-stroke predict reduced maximum velocity.

Now, one may of course claim that this study is about myosin V whereas the effects on velocity have primarily been seen with myosin II. However, there is evidence from different studies (e.g. Muretta et al vs Trivedi et al PNAS 2015a,b and Moretto et al 2022 vs Llinas et al 2015) that the Pi-release mechanism is similar in different myosin. Indeed, also Debold and co-workers initially developed their model for myosin II whereas then applying it (or an easily recognizable version of it) to myosin V. Finally, in the present manuscript they add to this notion by claiming "load sensitivity of Pi-rebinding to the nucleotide-binding site may be conserved across several members of the myosin family."

In view of the above, I think that the least one can require is that the authors discuss how the idea of Pi-release after the power-stroke can be made consistent with a high maximum gliding velocity (of myosin II or V) that is also (largely) [Pi] insensitive (at least without velocity decreasing with increased [Pi]).

We dealt with this issue, as indicated in the prior response to reviews, by introducing a model in which myosin II can detach from actin in a post-powerstroke state (see point 1.F). See also Debold et al. 2011 and Debold et al. 2013 in which we modelled effects of Pi on velocity in the motility assay.

We have also added a section in the Discussion of this 2nd revision of the manuscript to address this issue (see lines 290-297).

2. In the section "Implications for transduction"

Lines 333-335: "Therefore, the rebinding of Pi to the active site should prevent the powerstroke and thus prevent or inhibit displacements as load is increased, because Pi will more readily rebind to the active site at higher resistive loads."

This may be incorrectly phrased. An alternative that, as far as I see, cannot be excluded is that Pi may bind with equal probability (rate constant) at different loads but only leads to detachment at high load. A mechanism of that type was proposed by Moretto et al (2022) but it seems to me, unless I have misunderstood something, that the authors have misinterpreted that model. Thus, in the model of Moretto et al., due to equilibrium between Pi at the second site and the active site (Fig. 4a of Moretto et al), allosteric effects of Pi-rebinding on actomyosin affinity is most likely mediated by occupation of the active site also in this case. The increased Pi-induced detachment rate at increased load per myosin head would follow in the multistep model of Moretto et al. from the free energy diagrams (extracted from their paper in figure 1, attached). In that model, the increased probability of detachment with increased load would be a lower difference in free energy between the detached MADPPi state and the AMADP state (or AMADPP' state) at higher load per myosin head (thick arrow; higher x-value in figure above). This follows from the location of the different free energy diagrams (for pre and post-powerstroke states) along the x-axis and the axis for free energy.

Our understanding, which reflects a widely accepted structural model of myosin's active site dynamics (see e.g. Sweeney and Houdusse 2010 or Llinas et al. 2015), is that if Pi returns to the active site it rapidly induces detachment from actin. So we believe that if Pi rebinds to myosin and does not lead to detachment it did not gain access to the active site. This view is based on structural and functional observations in myosin II (e.g. Dantzig et al. 1992) and myosin Va and VI (Llinas et al. 2015) where the authors concluded that at low load Pi is very unlikely to return to the active site. And once there it would induce detachment, which is consistent with our reductions in event lifetimes (Fig. 2). So it is difficult for us to see how Pi could rebind to myosin's active site at low load and not induce detachment.

Minor

Lines 318-319 Statement "This seems the more likely explanation given that the substitution is in switch I."

It may be appropriate to mention that Llinas et al, (2015) who also observed effects of this mutation (together with several others) on Pi-release came up with another interpretation."

We reference the first authors to describe this mutation, and also those who published the first structure revealing this residue. And we acknowledge the different views regarding Pi's exit and re-entry into the active site. Given this information, the current language as stated in the manuscript seems appropriate to us.

Reviewer #2 (Remarks to the Author):

This is a revised manuscript in which the authors have address each of the issues raised in my original review.

Thank you for making it a stronger manuscript.

Reviewer #3 (Remarks to the Author):

The authors addressed all my comments. The new data and analysis greatly enhanced the manuscript and I suggest to publish it in Nature Communications. There is just a minor issue that I suggest to fix before publication:

- In the revised manuscript the detachment rate plotted in Fig.4b is obtained from the time spent at peak force T_{pf} , as shown in Fig.4a. It is not clear from the figure and the caption how T_{pf} is determined, since the beginning of T_{pf} might be somehow arbitrary. Have the authors used some kind of threshold to set the beginning of T_{pf} ? I suggest to add a paragraph in the methods to explain this point.

We have added this as suggested (see lines 449-453). And thank you for suggesting this analysis it makes the findings more impactful.